# Clonal associations between lymphocyte subsets and functional states in rheumatoid arthritis synovium

Garrett Dunlap [1,39], Aaron Wagner [2,39], Nida Meednu [3,39], Ruoqiao Wang [4], Fan Zhang [1,5,6,7,8,9], Jabea Cyril Ekabe[3], Anna Helena Jonsson [1], Kevin Wei [1], Saori Sakaue [1,5,6,7,8], Aparna Nathan [1,5,6,7,8], Accelerating Medicines Partnership Program: Rheumatoid Arthritis and Systemic Lupus Erythematosus (AMP RA/SLE) Network*, Vivian P. Bykerk[10,11], Laura T. Donlin [10,11], Susan M. Goodman[10,11], Gary S. Firestein[12], David L. Boyle[12], V. Michael Holers [13], Larry W. Moreland[13,14], Darren Tabechian[3], Costantino Pitzalis [15,16,17], Andrew Filer [18,19,20], Soumya Raychaudhuri [1,5,6,7,8], Michael B. Brenner [1], Juilee Thakar [2,4], Andrew McDavid[2,40], Deepak A. Rao [1,40] ✉ & Jennifer H. Anolik [3,4,40] ✉

Rheumatoid arthritis (RA) is an autoimmune disease involving antigen-specific T and B cells. Here, we perform single-cell RNA and repertoire sequencing on paired synovial tissue and blood samples from 12 seropositive RA patients. We identify clonally expanded CD4 + T cells, including CCL5+ cells and T peripheral helper (Tph) cells, which show a prominent transcriptomic signature of recent activation and effector function. CD8 + T cells show higher oligoclonality than CD4 + T cells, with the largest synovial clones enriched in GZMK+ cells. CD8 + T cells with possibly virus-reactive TCRs are distributed across transcriptomic clusters. In the B cell compartment, NR4A1+ activated B cells, and plasma cells are enriched in the synovium and demonstrate substantial clonal expansion. We identify synovial plasma cells that share BCRs with synovial ABC, memory, and activated B cells. Receptor-ligand analysis predicted IFNG and TNFRSF members as mediators of synovial Tph-B cell interactions. Together, these results reveal clonal relationships between functionally distinct lymphocyte populations that infiltrate the synovium of patients with RA.

Synovial inflammation in rheumatoid arthritis (RA) involves a complex set of interactions between immune and non-immune cell subsets. A core feature of the immune response in seropositive RA is an adaptive immune response against citrullinated proteins involving both antigen-specific B cells and T cells[1] The activation of B cells in RA has long been appreciated, given the characteristic production of disease-associated autoantibodies, including rheumatoid factor (RF) and anti-cyclic-citrullinated peptide (anti-CCP) antibodies[2–5]. B cells may be activated locally within RA synovium, as studies on synovial tissue have provided evidence of somatic hypermutation (SHM) and clonal expansion[6,7]. Furthermore, CCP- and RF-specific B cells may undergo distinct activation pathways[8]. Synovial B cells can contribute antibody-independent functions as well, including antigen presentation and cytokine secretion, which involve interactions with other cell types[9–12].

A full list of affiliations appears at the end of the paper. *A list of authors and their affiliations appears at the end of the paper.
✉e-mail: darao@bwh.harvard.edu; jennifer_anolik@urmc.rochester.edu

Populations of T cells have likewise been strongly implicated in the initiation and maintenance of synovial inflammation in RA[13]. Genetic associations indicate a critical role for antigen presentation to CD4 + T cells via MHC class II in the development of RA[14,15]. Cellular profiling studies of RA synovial tissue and fluid have highlighted a large population of T peripheral helper (Tph) cells, as well as T follicular helper (Tfh) cells, both of which provide help to B cells through the production of IL-21 and CD40L[16–19]. Tph cells differ from Tfh cells in their migratory patterns, expressing chemokine receptors such as CCR2 and CCR5 to home to sites of peripheral inflammation such as the rheumatoid joint[20,21]. Large populations of CD8 + T cells also accumulate within RA synovium, including a prominent granzyme K-expressing population, which may contribute to synovial inflammation through inflammatory cytokine production rather than cytotoxicity[22].

Complementing studies of specific subsets of immune cells, a holistic picture of both immune and non-immune populations in RA is emerging through single-cell RNA-sequencing (scRNA-seq) atlases of synovial tissue samples[23–26]. These studies have highlighted the diversity of cell states present in the inflamed tissue of these patients, as well as how the presence and effects of these states may differ among patient subpopulations[25,26]. For lymphocytes, analyses of the T cell receptors (TCR) or B cell receptors (BCR) can provide unique insights into the expansion and developmental relationships of lymphocyte subsets, leveraging the feature that each new lymphocyte generates a unique TCR/BCR that is shared with its progeny. Further, BCR genes undergo somatic hypermutation during antigen selection which can reveal B cell developmental lineages. Studies tracking TCRs across tissues or longitudinally in RA patients have identified shared T cell clones in different joints[27,28], clonal expansion of specific cell subsets[29], persistence of expanded clones over time[30], and overrepresented gene rearrangements that may suggest shared antigenic targets[28,31,32]; however, reactivity of expanded TCRs from RA synovial CD4 + T cells to citrullinated peptides has been difficult to demonstrate[33]. Studies of BCR repertoires of RA patients have indicated somatic hypermutation in RA synovial B cells and identified potential specificities across B cells collected from synovial tissue or fluid[6,34–36]. A comprehensive examination of both the TCR and BCR repertoires of synovial tissue lymphocyte populations and across tissue and blood at the single-cell level has not been described. Such studies have the potential to directly link clonal features to the functional roles, developmental relationships, and cell–cell interactions of specific lymphocyte phenotypes, as has been achieved in studies of cancer immunotherapy and infectious disease[37–40].

Here, we use 5′ droplet-based scRNA-seq on T and B cells of synovial tissue and matched peripheral blood samples from 12 patients with RA to simultaneously study their transcriptomes and antigen receptor repertoires. Our study provides a high-resolution landscape of the clonal relationships within and between cell states, and further between inflamed synovial tissue and peripheral blood.

## Results

### Single-cell profiling of synovial tissue and peripheral blood lymphocytes

We collected synovial tissue ($n = 12$) and matched peripheral blood ($n = 10$) from individuals with RA that comprised a subset of a larger cohort analyzed as part of the Accelerating Medicines Partnership Program: Rheumatoid Arthritis and Systemic Lupus Erythematosus (AMP RA/SLE) Network, prioritizing samples with high synovial cell yields and evident lymphocyte populations by flow cytometry[26]. Donors had a mean age of 63.6 years (range 28–80) and were predominantly female ($n = 11$). Patients in the cohort had a Clinical Disease Activity Index (CDAI) classification of moderate or high with a mean score of 26.4 (range 10.5–64.0), and were mainly lymphoid in phenotype, but otherwise had a range of disease duration, treatment, and

cell-type abundance phenotype (CTAP) (Supplementary Fig. 1A and Supplementary Data 1)[26,41,42].

Tissue samples were previously disaggregated into single-cell suspensions for an unbiased analysis of the cell states present in RA synovium using RNA and cell-surface protein profiling[26]. Cryopreserved synovial cells remaining after the initial analysis were thawed and sorted to isolate CD45 + CD3+ and CD45 + CD19+ populations, which were subsequently encapsulated into droplets and used to generate gene expression, cell-surface protein, and TCR/BCR single-cell sequencing libraries (Fig. 1A and Supplementary Fig. 1B). Sorted CD3 + T cells and CD19 + B cells were loaded in order to maximize the numbers of both cell lineages captured (Fig. 1E) and minimize sample bias based on varying lymphocyte abundance (Supplementary Fig. 1C, D). Cryopreserved PBMCs were thawed and sorted in parallel. Following a unified single-cell analysis pipeline of all samples, we recovered a total of 84,750 cells. After applying QC criteria, 83,159 cells remained, which we used to perform an initial round of unsupervised clustering at low resolution (Fig. 1B and Supplementary Fig. 2). On average, we obtained 2888 cells per synovial tissue sample (range 804–5188) and 4851 cells per blood sample (range 3442–6601) (Fig. 1C and Supplementary Fig. 1A).

Through parallel examination of gene expression and protein detection, we identified several populations of T cells expressing markers such as CD3E and IL7R, as well as a cluster containing B and plasma cells expressing CD79A and MS4A1 (CD20) (Fig. 1D and Supplementary Fig. 3A, B). Cells from both lineages could be found in all samples (Fig. 1E, F). We further identified a cluster containing proliferating T and B cells, characterized by the expression of markers including MKI67 and TYMS. Lastly, we identified two rare populations of contaminating cells expressing markers of fibroblasts (PRG4 and FN1) and monocytes (S100A8 and LYZ), which were excluded from subsequent analyses (Fig. 1D and Supplementary Fig. 3A, B).

Of the 81,708 lymphocytes captured across all samples, we obtained paired TCR or BCR information for 73,185 cells. In the synovial tissue, an average of 84.7% of lymphocytes per sample had an associated TCR or BCR (range 72.9–96.1%), while in the blood, 93.1% of lymphocytes per sample had this information on average (range 89.3–96.8%) (Fig. 1G and Supplementary Fig. 3C).

### Differential abundances of CD4 + T cell populations in synovial tissue and blood

Subclustering of the 35,301 CD4 + T cells obtained from synovial tissue and blood samples identified 14 CD4 + T cell subsets (Fig. 2A, B, Supplementary Fig. 4A, and Supplementary Data 2). We identified two clusters expressing the B cell chemoattractant CXCL13, one of which was composed solely of Tph cells, while the other included some cells with detectable CXCR5, suggesting the cluster contained a mixture of Tfh and Tph cells (Supplementary Fig. 4B, C)[16,24,26]. Both clusters expressed genes associated with B cell help, such as MAF, and both scored highly for a Tfh cell gene signature (Supplementary Fig. 4B, D and Supplementary Data 3). While the Tfh/Tph cluster displayed higher expression of markers such as IL7R and CD69 compared to the Tph cluster, the Tph cluster had significantly elevated expression of PDCD1, CTLA4, LAG3, and others. Further, the pure Tph subset produced higher gene scores of TCR signaling and T cell activation compared to other CD4+ clusters, suggesting that the cells in this cluster are activated (Fig. 2C, Supplementary Fig. 4D, E, and Supplementary Data 3).

Several other clusters of memory cells with distinctive expression of chemokines, chemokine receptors, and granzymes were present, including CCR7+ (which may represent central memory cells), IL7R+ CCL5+, GZMA+CCL5+, and GZMK+ populations. In addition, we identified a population of cytotoxic CD4+T cells, marked by the expression of GNLY, PRF1, and GZMB, as well as a cluster characterized by a strong interferon-response signature. A small population likely containing a

mix of naive and memory cells was distinguished primarily by increased expression of members of the GIMAP family (e.g., *GIMAP4, GIMAP5*), which has been associated with survival and quiescence in lymphocytes (Supplementary Data 2)[43].

Subclustering also revealed two populations of regulatory T cells (Tregs). Both expressed *FOXP3, CTLA4,* and *TIGIT,* though one population was marked by stronger expression of *IL2RA* (encoding CD25)

and IL32, while the other displayed higher *CCR7* and *TCF7*, suggesting the presence of populations of effector and central memory-like Tregs, respectively[44]. A large cluster containing naive T cells expressing *SELL, TCF7,* and *CCR7* displayed the strongest expression signature for a previously identified naive T cell gene set[45] (Supplementary Fig. 4B–E). Lastly, an actively proliferating cluster of cells, as well as a subset with elevated mitochondrially encoded RNAs, could be detected

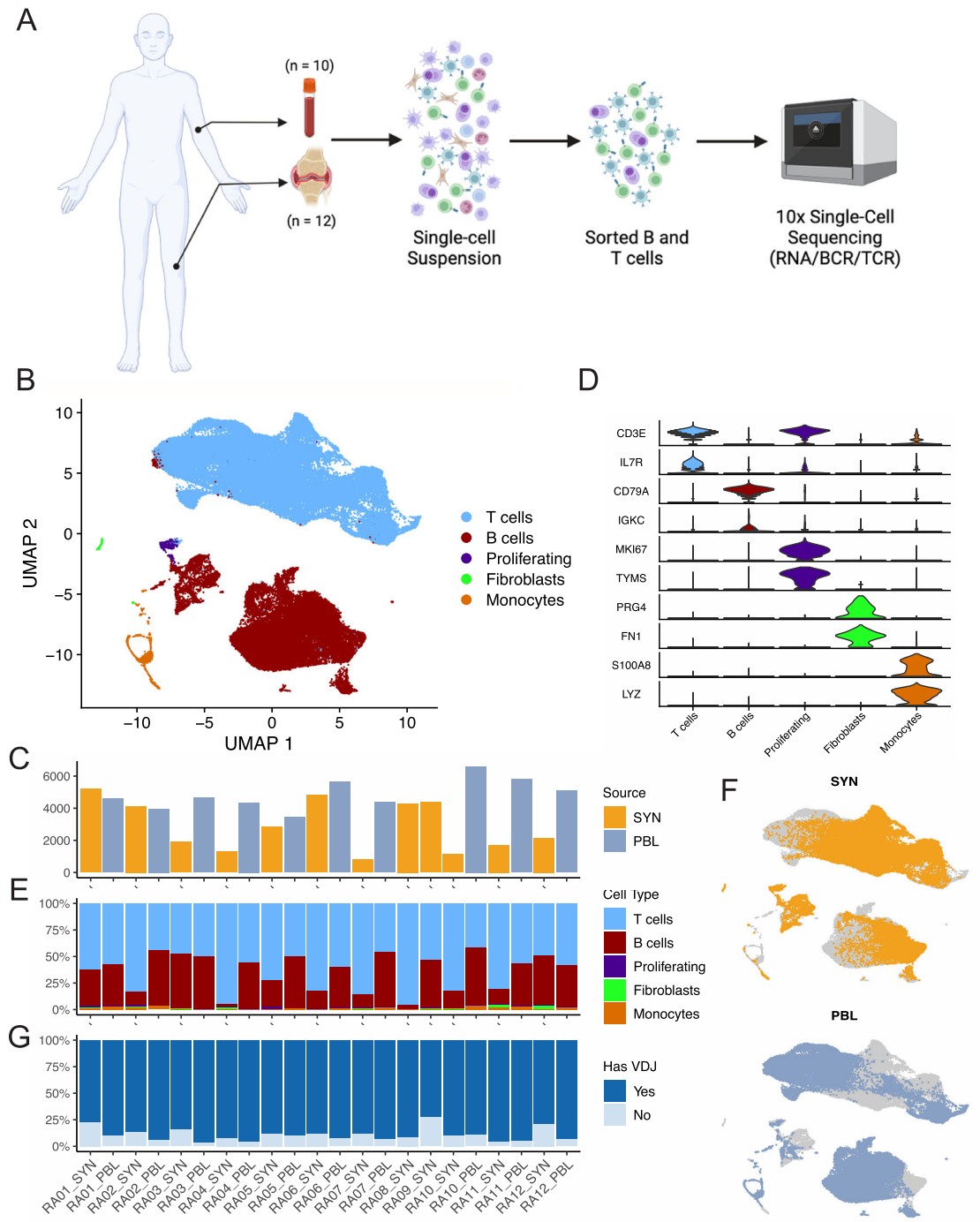

**Fig. 1 | Sorting and single-cell analysis of matched blood and synovial tissue T cells and B cells. A** Schematic showing the overall study design. T and B cells were isolated from synovial tissue biopsies (*n* = 12) and matched peripheral blood (*n* = 10). Single-cell libraries for 5′ gene expression and receptor repertoires were generated using the 10X Genomics platform. **B** Unsupervised clustering and UMAP projection of 83,159 cells that passed QC. **C** Bar plot of the number of cells recovered for each sample, color denotes tissue origin. **D** Violin plots showing the expression distribution of select markers for each identified cell population. **E** Bar plot of the cluster composition for each sample. **F** UMAPs of the combined clustering, separated by synovial tissue (top) and peripheral blood (bottom). **G** Bar plot highlighting the recovery of VDJ information for each sample, excluding fibroblast and monocyte populations. PBL peripheral blood lymphocytes, SYN synovial tissue. Figure 1A, created with BioRender.com, released under a Creative Commons Attribution-NonCommercial-NoDerivs 4.0 International license.

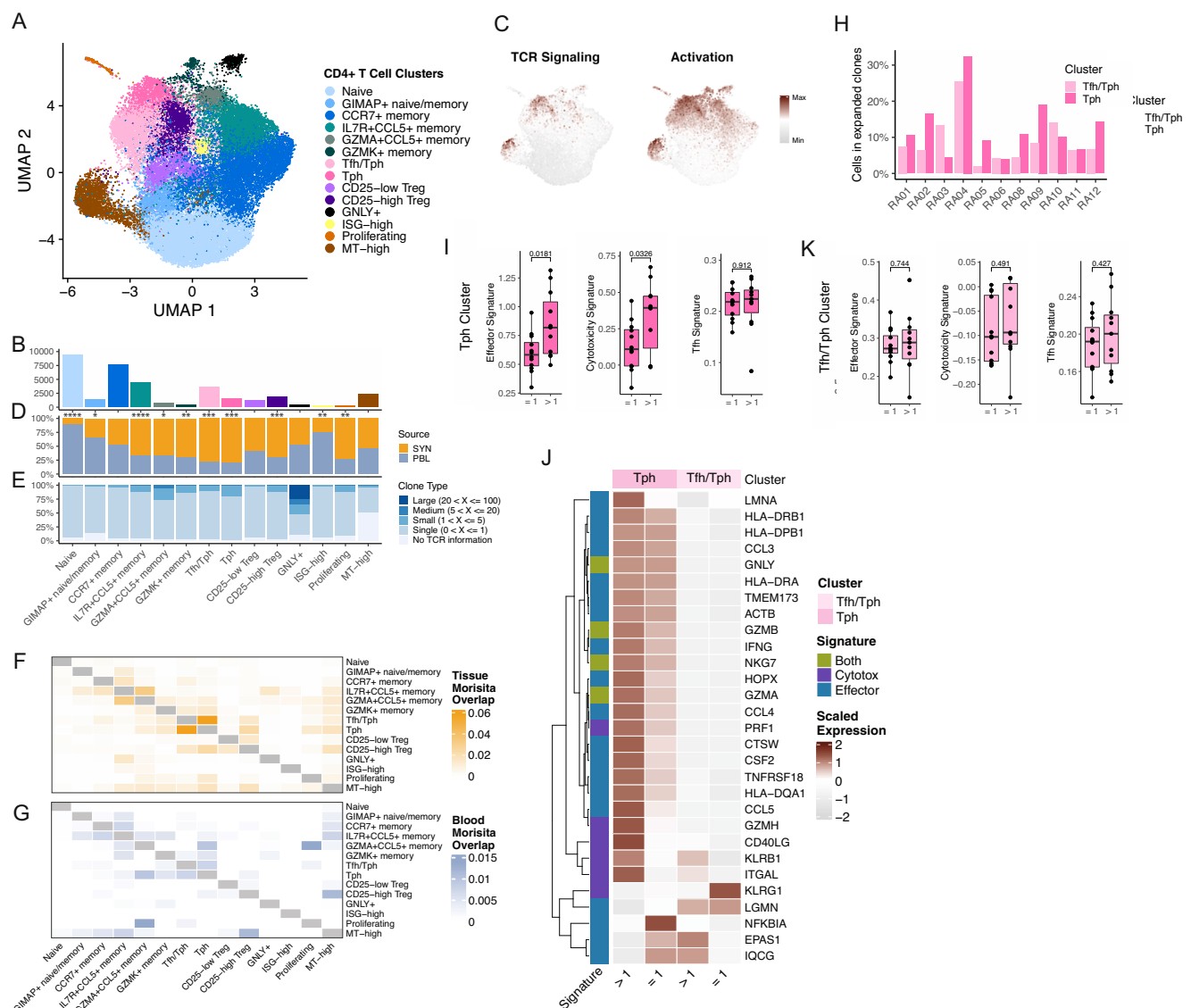

**Fig. 2 | Differential abundances of CD4 + T cell populations in synovial tissue and blood. A** UMAP projection of CD4 + T cell reclustering. **B** Bar plot of the number of cells included in each cluster. **C** UMAPs of TCR signaling (left) and activation (right) signatures. **D** Bar plot of the tissue distribution for cells in each cluster, with significance determined by two-sided paired *T*-tests (Naive, *p* = 2.22E-08; GIMAP+ naive/memory, *p* = 0.0202; CCR7+ memory, *p* = 0.886; IL7R + CCL5+ memory, *p* = 2.03E-05; GZMA + CCL5+ memory, *p* = 0.329; GZMK+ memory, *p* = 0.00571; Tfh/Tph, *p* = 0.000125; Tph, *p* = 0.000666; CD25-low Treg, *p* = 0.0758; CD25-high Treg, *p* = 0.000205; GNLY+, *p* = 0.808; ISG-high, *p* = 0.00164; Pro-liferating, *p* = 0.00332; MT-high, *p* = 0.19). **E** Bar plot of the clone size distribution for each cluster. **F**, **G** Heat map of pairwise clonal overlap values calculated using Morisita's index for synovial tissue (**F**) and blood (**G**). **H** Bar plots of the percent of synovial tissue Tph and Tfh/Tph cells found to be expanded, per donor. **I** Box plots of the effector signature (left), cytotoxicity signature (middle), and Tfh signature (right) distribution for expanded and non-expanded Tph cells. Each dot represents a donor (*n* = 12), and *P* values were determined by paired *T*-tests. **J** Heat map of the average expression of effector and cytotoxicity signature genes, comparing expanded and non-expanded Tph and Tfh/Tph cells. **K** Box plots of the effector signature (left), cytotoxicity signature (middle), and Tfh signature (right) dis-tribution for expanded and non-expanded Tfh/Tph cells. Each dot represents a donor (*n* = 12), and *P* values were determined by paired *T*-tests. For **I** and **K**, box plots are defined with lower and upper edges corresponding to the first and third quartiles (the 25th and 75th percentiles), respectively, the upper whisker extends from the 75th percentile to the largest value no further than 1.5x the interquartile range (IQR) from the edge, the lower whisker extends from the edge to the smallest value at most 1.5x the IQR of the lower edge, and the median is the center line.

(Supplementary Fig. 4B–E). In total, T cell populations identified by this present clustering analysis were well aligned with those identified in analyses of a larger set of synovial tissue samples (of which the 12 synovial samples studied here are a subset), where clustering was performed through a combination of RNA and cell-surface protein profiling, (Supplementary Fig. 4F)[26].

We next identified T cell subsets that had differential repre-sentation in either synovial tissue or peripheral blood. Several memory/effector cell populations were enriched in synovium compared to blood, including the Tph, Tfh/Tph, CD25-high Treg,

*IL7R+CCL5+* memory, *GZMA+CCL5+* memory, *GZMK+* memory, and proliferating clusters (Fig. 2D and Supplementary Fig. 4G). In con-trast, blood samples contained increased abundances of the naive, GIMAP+, and interferon-stimulated clusters. On average, 45% of the CD4+ population for each blood sample was composed of cells from the naive cluster (range 18–66%), while only 5% of synovial tissue CD4 + T cells had a naive phenotype (range 2–15%). Thus, a range of memory/effector T cell populations with distinct tran-scriptomic signatures are enriched in RA synovium compared to blood.

## Clonally expanded Tph cells display increased effector and cytotoxic signatures

We then sought to examine the TCR repertoire of the CD4+ populations across synovial tissue and blood. Among the CD4+ populations, the *GNLY*+ cytotoxic cluster displayed the highest clonal expansion and comprised nearly all clones larger than 20 cells. Though detected clone size can be impacted by the number of cells analyzed, it is notable that the *GNLY*+ cluster was of lower abundance compared to other CD4 clusters (Fig. 2B), and clonal expansion was still detectable. Clonal expansion, defined as two or more cells with an identical TCR, was also identified among the Tph and *GZMA + CCL5*+ memory clusters (Fig. 2E and Supplementary Fig. 4H, I). An analysis of clonal sharing among the synovial tissue clusters revealed a high degree of clonal overlap between the Tph and Tfh/Tph clusters, suggesting that cells in these clusters are developmentally related despite their transcriptomic differences (Fig. 2F, G). By subdividing the Tfh/Tph cluster into cells with and without detectable CXCR5 transcript, we identified a set of clones represented in both CXCR5+ Tfh cells and cells within the Tph cluster (Supplementary Fig. 4J). In addition, the synovial Tph cluster shared the highest clonal overlap with the cycling cluster compared to all other subsets, highlighting the active proliferation of the Tph population within the synovial tissue. We also identified a small number of T cells from blood with TCRs that matched synovial Tph or Tfh/Tph cells; these blood T cells were most frequently in the blood Tph or Tph/Tfh clusters (Supplementary Fig. 4K). The pattern of clonal sharing across blood CD4 + T cell clusters overall appeared generally similar to that in synovium, but with a lower degree of overlap, and with the *GZMA + CCL5*+ subset showing the highest clonal overlap with the proliferating blood CD4 + T cells (Fig. 2F, G).

After identifying the Tph population as highly represented in the synovial tissue, clonally expanded, and actively proliferating, we sought to identify features that differentiate clonally expanded and unexpanded Tph cells in the tissue. Across donors, an average of 12.5% of Tph cells belonged to an expanded TCR clone (range 4–32.5%) (Fig. 2H). By examining transcriptional differences between these cells and Tph cells belonging to an unexpanded clone, clonally expanded Tph cells showed significantly increased expression of signatures of effector function and cytotoxicity, including elevated expression of *IFNG, PRF1, CD40LG*, and *CCL5* (Fig. 2I, J). In contrast, no difference in the Tfh gene signature score was present between expanded and unexpanded cells, suggesting that clonally expanded Tph cells do not lose their B cell-helping functions. Interestingly, a similar comparison using cells from the Tfh/Tph cluster yielded no significant differences between expanded and unexpanded cells, further suggesting a unique set of features among cells in the Tph clusters (Fig. 2K).

## Expanded CD8 + T cell clones across tissue types differ by patterns of *GZMK* and *GZMB* expression

Next, we isolated and subclustered the CD8 + T cells present in the synovial tissue and blood samples, which revealed nine CD8 + T cell populations that could be found across samples (Fig. 3A, B, Supplementary Fig. 5A, and Supplementary Data 2). Among these, three populations were distinguished by expression patterns of *GZMK* and *GZMB*, all of which had strong corresponding matches to clusters identified in a recent study of synovial tissue (Supplementary Fig. 5B, C)[26]. One of these clusters solely expressed *GZMK*, while another expressed GZMK and low levels of *GZMB*. Both of these populations also had elevated *GZMA* and *CCL5*, while the GZMK/B+ cluster was further differentiated through increased expression of markers, including *CCL4* and *HLA-DRA*, and decreased *IL7R*. The third population, characterized by expression of *GZMB* only, appeared highly cytotoxic through gene module analysis and elevated expression of *GNLY* and *PRF1* (Supplementary Fig. 5B–D). We also isolated a

population of likely resident memory CD8 + T (Trm) cells, characterized by increased *ZNF683* and *XCL1*, which was supported more broadly through examination of a previously published Trm gene list and consistent with a recent description of synovial Trm cells (Supplementary Data 3)[46,47]. We identified one naive CD8+ population with high expression of *SELL* and *LEF1* and a naive gene module signature (Supplementary Fig. 5B–D). Another population of likely naive cells was also identified, characterized by increased expression of *IGTB1* and *LMNA*. Lastly, we found separate clusters of CD8 + T cells with elevated interferon-response, proliferation, or mitochondrial gene modules (Supplementary Fig. 5B–D).

An analysis of cluster representation between synovial tissue and peripheral blood samples found the *GZMK/B*+ cluster to be highly increased in synovial tissue samples (Fig. 3C and Supplementary Fig. 5F), consistent with a recent report[22]. Clusters solely expressing *GZMK* or *GZMB* were not significantly different in abundance between tissue compartments, though the *GZMK*+ memory population trended higher in synovial tissue, while the *GZMB* $T_{EMRA}$ cluster appeared slightly elevated in the blood (Supplementary Fig. 5F). Additionally, the Trm, proliferating, and mitochondrial-high clusters had elevated representation in synovial tissue samples. In contrast, the naive and *ITGB1*-elevated populations could both be found at higher frequencies in the blood (Fig. 3C and Supplementary Fig. 5F).

After characterizing the cell states present among the CD8+ population, we sought to connect these clusters' clonal attributes. Broadly, we found a much larger degree of clonal expansion in the CD8+ compartment compared to the CD4+ subsets. We found the strongest clonal expansion among the *GZMK/B*+ memory and *GZMB* + $T_{EMRA}$ clusters, and also noted cells in the ISG-high and proliferating clusters to belong to expanded clones (Fig. 3D and Supplementary Fig. 5G). Comparing the top 50 largest clones across the synovial tissue and blood revealed striking differences in cluster composition. Cells from the top 50 clones in the synovial tissue belonged overwhelmingly to the *GZMK/B*+ memory population. By contrast, the top 50 clones in the blood belonged to the *GZMB* + $T_{EMRA}$ cluster, with minor representation of other populations (Fig. 3E, F). This analysis of the largest clones, and a broader examination of the clonal overlap between subclusters in either tissue, showed sharing between the *GZMK/B*+ memory cluster and proliferating cells of the synovial tissue, while the *GZMB* + $T_{EMRA}$ was found to overlap with the proliferating component of the blood (Supplementary Fig. 5H, I). To further analyze this, we divided the proliferating cluster into subsets and mapped these cells onto the other CD8+ cells of this dataset. Compared to the general cluster proportions, the tissue and blood proliferating cells mapped heavily to the *GZMK/B*+ (201/261 tissue proliferating cells) and *GZMB* + $T_{EMRA}$ (24/38 blood proliferating cells) clusters, respectively, suggesting an active, though likely differing, role of these two populations in their distinct tissues (Fig. 3G).

A further examination of clonal overlap across tissues revealed a small degree of sharing between the tissue and blood components of either the *GZMK/B*+ memory or *GZMB* + $T_{EMRA}$ clusters separately, but clonal overlap between synovial tissue GZMK/B+ cells and blood *GZMB* + $T_{EMRA}$ cells was essentially absent (Supplementary Fig. 5J). Of the top 50 clones in both tissue compartments, only six could be found in both, nearly all of which were larger in the blood than the synovial tissue. Notably, four of these clones retained a *GZMB*+ skew even in the synovial tissue (Fig. 3E, F). Together, these results suggest functionally distinct roles for the expanded GZMK/B+ cells of the synovial tissue and the *GZMB* + $T_{EMRA}$ cells of the blood. The findings corroborate recent work suggesting that the *GZMK/B*+ cells likely do not arrive in the synovial tissue as *GZMB*+ cytotoxic T cells that subsequently alter their phenotype, and instead may enter the synovium as *GZMK*+ cells or expand locally in the tissue as they adopt a phenotype that includes *GZMK* expression[22].

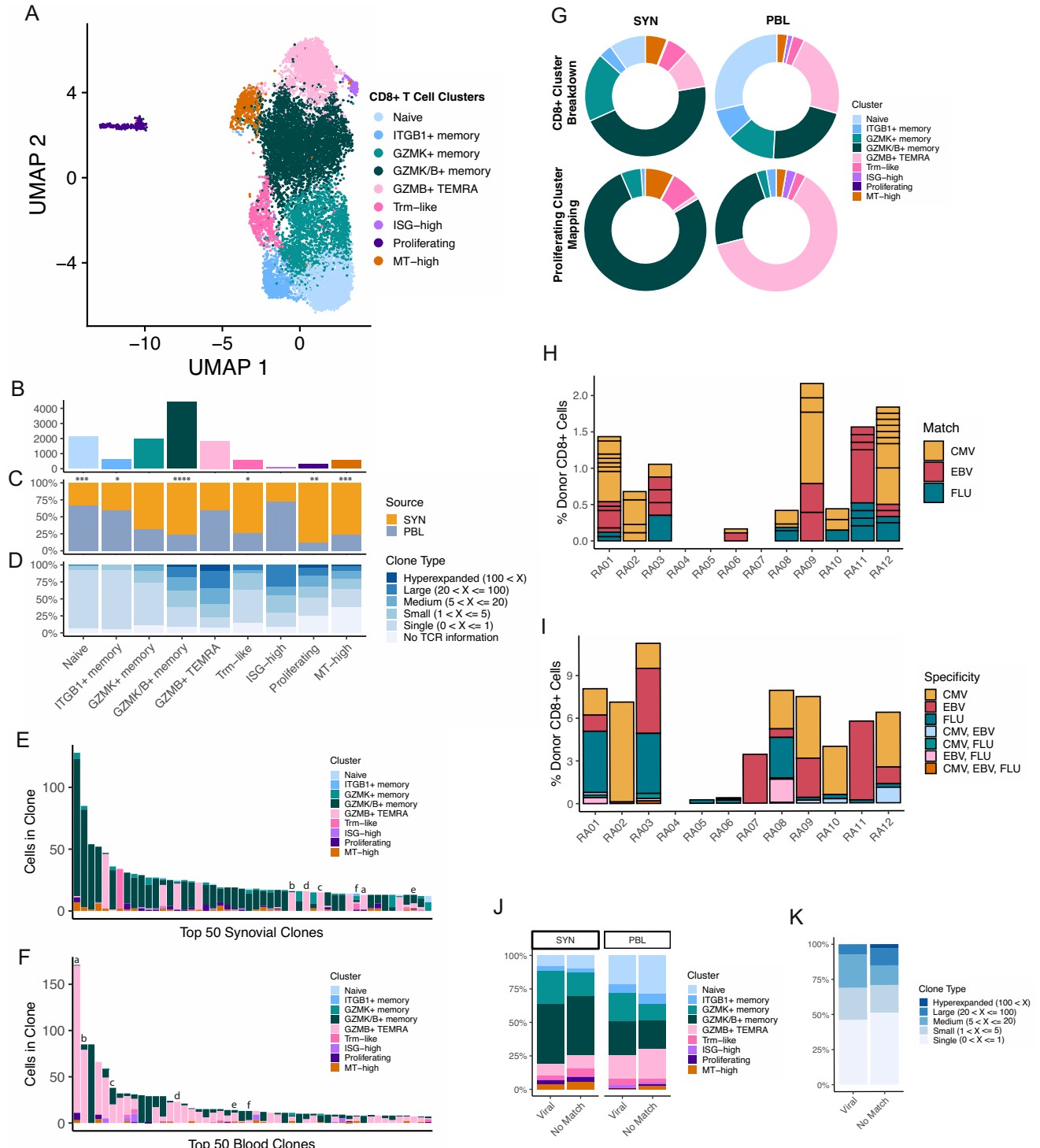

**Fig. 3 | GZMK and GZMB + T cells are not clonally related. A** UMAP projection of CD8 + T cell reclustering. **B** Bar plot of the number of cells included in each cluster. **C** Bar plot of the tissue distribution for cells in each cluster, with significance determined by two-sided paired *T*-tests (Naive, *p* = 0.000306; ITGB1+ memory, *p* = 0.04; GZMK+ memory, *p* = 0.0523; GZMK/B+ memory, *p* = 9.16E-05; GZMB + TEMRA, *p* = 0.148; Trm-like, *p* = 0.0473; ISG-high, *p* = 0.304; Proliferating, *p* = 0.00697; MT-high, *p* = 0.000885). **D** Bar plot of the clone size distribution for each cluster. **E, F** Bar plots of the clone sizes and cluster breakdowns of the top 50 most expanded clones for synovial tissue (**E**) and blood (**F**) separately. Letters

above bars denote clones shared between synovial tissue and blood. **G** Donut plots of the proliferation cluster mapping proportions compared to the general cluster proportions. **H** Bar plot of the percent of CD8+ cells that are exact viral-reactive matching, per patient, split by virus. Box size denotes the size of the clone. **I** Bar plot of the percent of CD8+ cells that have a GLIPH2 motif-matching viral-reactive clones, per patient, split by virus. **J** Bar plot of the cluster distribution for GLIPH2 motif viral-reactive matching and non-matching cells, split by tissue source. **K** Bar plot of the clone size distribution for GLIPH2 motif viral-reactive matching and non-matching cells.

## Predicted virus cross-reactive CD8 + T cells do not display altered expansion or phenotypic characteristics

Viral infection has long been connected with the potential for the development and maintenance of autoimmune diseases[48]. One possibility that may instigate this phenomenon is the presence of cross-reactive epitopes between a virus and endogenous proteins, which may drive the activation and expansion of a set of T cells. We sought to identify potentially viral-specific T cells in the synovial tissue to evaluate the extent of potential viral cross-reactivity among RA synovial CD8 T cells. To accomplish this, we gathered databases of previously identified CMV, EBV, and influenza A-specific T cells[49,50]. We then identified exact matches with the beta chain CDR3 sequence and HLA between the RA synovial CD8 + T cells analyzed here and those in the database (Supplementary Fig. 6A). Although three of the RA patients (RA04, RA05, and RA07) expressed less common HLA alleles (Supplementary Data 5), precluding the ability to find matches with previously-discovered specific clones, we were able to identify matching clones in the majority of patients (Fig. 3H and Supplementary Fig. 6B). Within and between patients, we further found these matches to be directed against multiple different epitopes for each of the viruses tested (Supplementary Fig. 6C). When examining which cluster these matching T cells belong to, we found a spread across clusters, with no cluster being significantly overrepresented compared to non-matching cells (Supplementary Fig. 6D). Further, we noted that few matching clones were expanded (16/56), with the largest matching clone comprising only seven cells. For comparison, analysis of CD8 + T cells from blood yielded similar results, with a comparable number of viral-reactive T cells, and a similarly broad distribution of cell phenotypes represented by viral-reactive cells (Fig. 3J and Supplementary Fig. 6E, F).

Requiring an exact match of clone sequence and HLA can provide strong evidence of the capacity for reactivity against a virus; however, identifying shared motifs between viral-specific T cells and those in our dataset may allow the identification of a larger set of T cells with the potential to detect viral epitopes. Thus, we employed GLIPH2 to identify motifs within the beta chain CDR3 sequence enriched in viral-specific T cells and our CD8 + T cells[51]. After running the GLIPH2 algorithm individually with each virus (CMV, EBV, and influenza) for RA synovial T cells, we filtered motifs that contained both virus and RA clones, and only those with an HLA match (Supplementary Fig. 6G). We obtained a larger number of clones belonging to a viral-associated GLIPH motif (369 unique clones) compared to our exact matches (56 unique clones), which was variable across donors (Fig. 3I and Supplementary Fig. 6H, I). To strengthen the association of the hits within our data, we sought to identify a relationship between patient age and the percent of CD8 + T cells associated with potential viral reactivity. Though not significantly correlated, we found trends between donor age and the percent of potential CMV- and EBV-reactive clones, consistent with the dynamics of the anti-viral repertoire with age[52,53] (Supplementary Fig. 6J). Similar to results with exact matches, a breakdown of the cluster makeup of motif-matching cells and non-matching cells again revealed no significant differences among synovial CD8 + T cells, with similar results also obtained for CD8 + T cells from blood (Fig. 3J). Further, we identified no differences in the clone size distributions between these two groups, and also found no clones belonging to viral GLIPH motifs among the 50 largest synovial tissue clones (Fig. 3K).

## Identification of activated innate T cell populations in RA synovium

Populations of innate T cells have previously been associated with RA, including natural killer (NK), γδ, and mucosal-associated invariant T (MAIT) cells, though clear roles for many of these subsets in RA remain elusive[54–57]. We identified and subclustered innate T cells in the synovial tissue and blood samples, resulting in seven subsets

representing multiple innate lineages (Fig. 4A, Supplementary Fig. 7A, B, and Supplementary Data 2). Two populations of γδ T cells, including a Vδ1 subset characterized by expression of *TRDV1* and a Vδ2 subset expressing *TRDV2* and *TRGV9* (the TCR γ variable gene commonly paired with TRDV2), were retrieved[58]. The Vδ1 population had an elevated expression of *GZMB* and *TIGIT*, while Vδ2 cells had higher levels of *TYROBP*. MAIT cells were also identified among the innate cells, expressing markers including *SLC4A10, AQP3*, and *ZBTB16* (Fig. 4B and Supplementary Fig. 7C). Two populations of NK cells were detected, including CD56-dim and CD56-bright subsets, that aligned with corresponding NK populations in a scRNA-seq reference (Supplementary Fig. 7D)[26]. These CD3- subsets were not intended to be included in the sorting scheme used in this study; thus, their frequency may not reflect the true representation in these samples. In addition to these clusters, we also found a population characterized by the expression of *ZNF683* (encoding Hobit), which appears to contain both γδ and NK cells and a population of innate cells with elevated mitochondrial gene expression (Supplementary Fig. 7C). When comparing the frequency of these populations between synovial tissue and peripheral blood, only the Vδ1 subset had significantly increased representation in the synovium, comprising a mean of 2.1% of T cells in the tissue (range 0.1–4.5%) and 0.5% (range 0.1–1.4%) in the blood (Fig. 4C and Supplementary Fig. 7E).

Leveraging the dataset's repertoire information, we confirmed the presence of MAIT cells through an examination of the TCR alpha chain rearrangements. The MAIT population was characterized by its use of TRAV1-2, often accompanied by TRAJ33, TRAJ20, or TRAJ12 rearrangements[59]. The pairing of the TRAV1-2 and TRAJ33/20/12 gene rearrangements could be detected in over half of the cells with an associated TCR in the cluster, while it was largely absent across other T cells in the dataset (Fig. 4D and Supplementary Fig. 7F, G). We then sought to clonally track these MAIT cells between tissue and blood and found expanded clones uniquely represented in either tissue, as well as a subset of clones that were present in both synovium and blood (Fig. 4E and Supplementary Fig. 7H).

To characterize transcriptional differences between the clonally expanded and non-expanded MAIT across these tissues, we examined signatures of activation. Synovial MAIT cells had elevated activation scores compared to MAIT cells in the blood (Fig. 4F). A similar pattern was observed with Vδ1 and Vδ2 cells (Supplementary Fig. 7I). Because MAIT cells are known to become activated through both TCR-dependent and -independent mechanisms[60], understanding how this relates to differences between tissues is relevant to better decipher the potential role of these cells in RA synovium. Gene signatures of TCR-dependent and -independent MAIT activation from multiple independent sources showed significantly enriched scores for both mechanisms in synovium compared to blood[61,62]. Both clonally expanded and non-expanded MAIT cells from synovial tissue had higher signatures for both mechanisms compared to their blood counterparts, but no significant difference could be detected between the expanded and non-expanded subsets of either tissue, suggesting subsets of MAIT cells in synovial tissue may become activated by either TCR-dependent and independent mechanisms (Fig. 4G, H, Supplementary Fig. 7J, K, and Supplementary Data 3). Together, these transcriptomic and repertoire data provide strong evidence for the presence of a defined MAIT cell population within RA synovium that appears activated.

## Activated B cells are enriched in the synovium

Analogous to the T cell analysis, we characterized 27,869 B cells through subclustering to obtain eight B cell and two plasma cell populations (Fig. 5A–C, Supplementary Fig. 8A–C, and Supplementary Data 2). We annotated four of these populations as naive subsets based on relatively higher expression of naive markers such as IgD and *TCL1A*, strong mapping to naive B cells from blood and tonsil in the

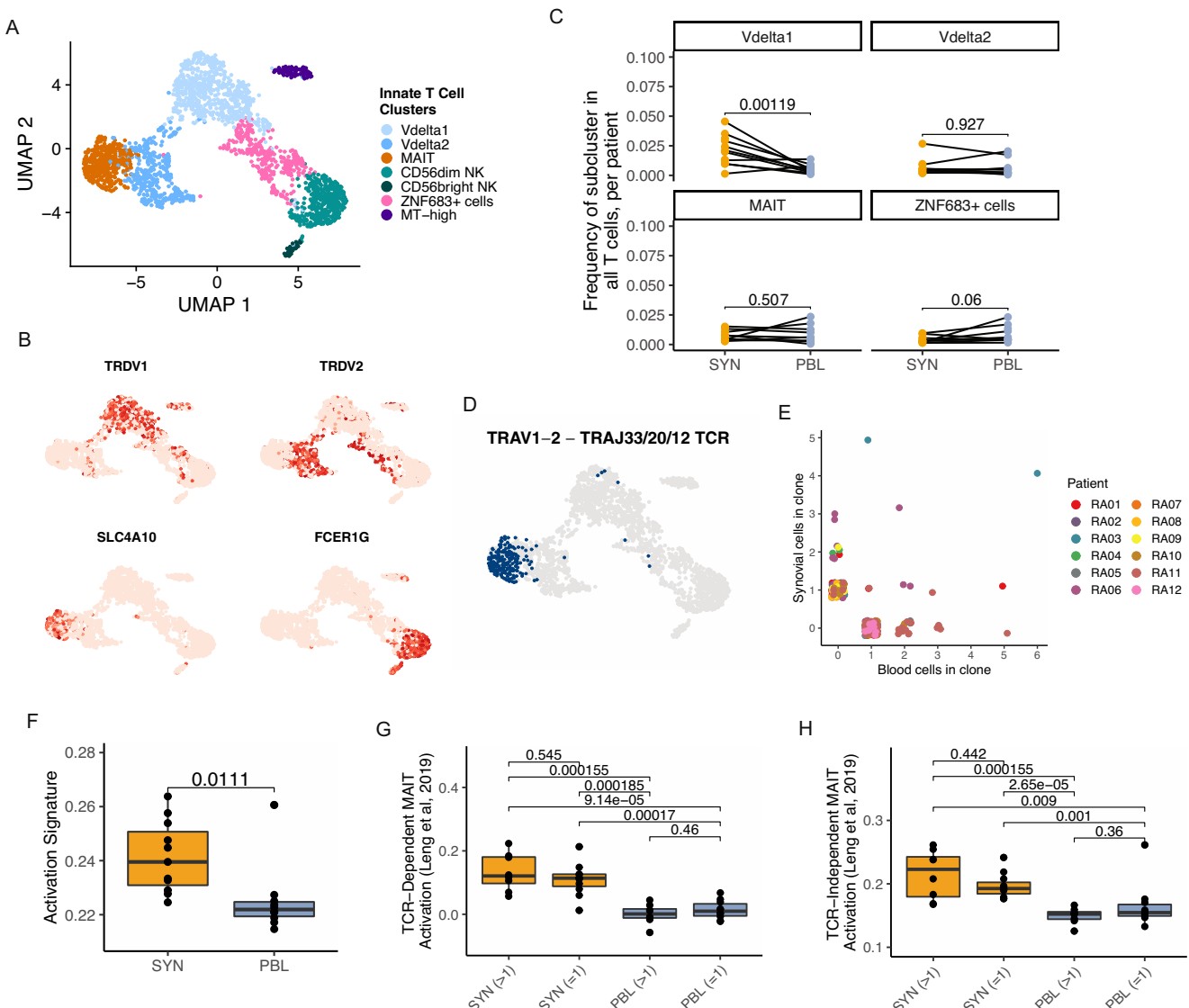

**Fig. 4 | Innate T cell populations have increased activation signatures in the synovium. A** UMAP projection of innate T cell reclustering. **B** UMAPs of the gene expression levels of select innate T cell lineage markers. **C** For each selected cluster, the frequency of cell representation is a proportion of all T cells in a sample. Each dot represents a single sample, and lines denote paired blood and tissue for a donor (*n* = 12 donors). Significance is determined by paired *T*-tests, with multiple testing corrections. **D** UMAP highlighting innate T cells with a TRAV1-2 and either TRAJ33, 20, or 12 gene rearrangement. **E** Scatter plot of the size of each MAIT cell clone across tissue sources, colored by the patient. **F** Box plot of an activation signature in

the MAIT cell cluster. **G, H** TCR-dependent (**G**) and independent (**H**) MAIT cell activation signatures from Leng et al, 2019, split by tissue and clonal expansion. Each dot represents a single sample (*n* = 12). Significance was calculated using paired Wilcox testing with Holm correction. For **F–H**, box plots are defined with lower and upper edges corresponding to the first and third quartiles (the 25th and 75th percentiles), respectively, the upper whisker extends from the 75th percentile to the largest value no further than 1.5x the interquartile range (IQR) from the edge, the lower whisker extends from the edge to the smallest value at most 1.5x the IQR of the lower edge, and the median is the center line.

published literature (Fig. 5E) and recently identified in RA synovial tissue using RNA and surface protein expression (Supplementary Fig. 8B)[26]. Two of the naive B clusters were distinguished by relative levels of IgD expression (Naive-IgD-low, Naive-IgD-high). The Naive-IgD-low is also distinguished by higher *FCER2* (CD23). The other two naive clusters had elevated expression of *HSPA1B* (Naive-*HSPA1B*+) or higher mapped mitochondrial reads (Naive-MT-high) and had weaker mapping to a naive B cell state (Fig. 5E and Supplementary Fig. 8B, D). These latter two clusters were the dominant naive population in the synovium. Naive-*HSPA1B*+ has an activated phenotype with upregulation of *NR4A1* and *DNAJB1* in addition to *HSPA1B*. Given the upregulation of *ZEB2* and *ITGAX* (CD11c) (Supplementary Fig. 8D, E), this subset resembles an activated naive B cell described as expanded in the blood of patients with lupus[63]. The other six clusters are non-naive and dominate the synovial B cells. On average, 64.5% of the B cells for each

blood sample were composed of cells from the naive clusters (IQR 59.5–86.7%), while only 9.1% of synovial tissue B cells had a naive phenotype (IQR 5.5–12.0%). We identified a cluster of memory B cells expressing memory markers, including *CD27*, *TNFRSF13B*, *S100A10*, and *S100A4* (Fig. 5B and Supplementary Fig. 8C–E). This memory cluster maps to a peripheral blood memory B cell signature (Fig. 5E) and the switched memory population described by ref. 26 (Supplementary Fig. 8B). Finally, over 60% of blood and over 75% of synovial memory B cells are class-switched based on V(D)J sequencing (Fig. 5H).

Additional B cell populations included age-associated B cells (ABCs), activated-, and *LILRA4 + B* cells. Age-associated B cells (ABCs) expressed the canonical markers CD11c and *FCRL5* as previously described (Fig. 5B and Supplementary Fig. 8C)[24,26,63,64], as well as other ABC makers such as *TBX21* (T-bet*) and *ZEB2* (Fig. 5B and Supplementary Fig. 8D). Interestingly, one of the top

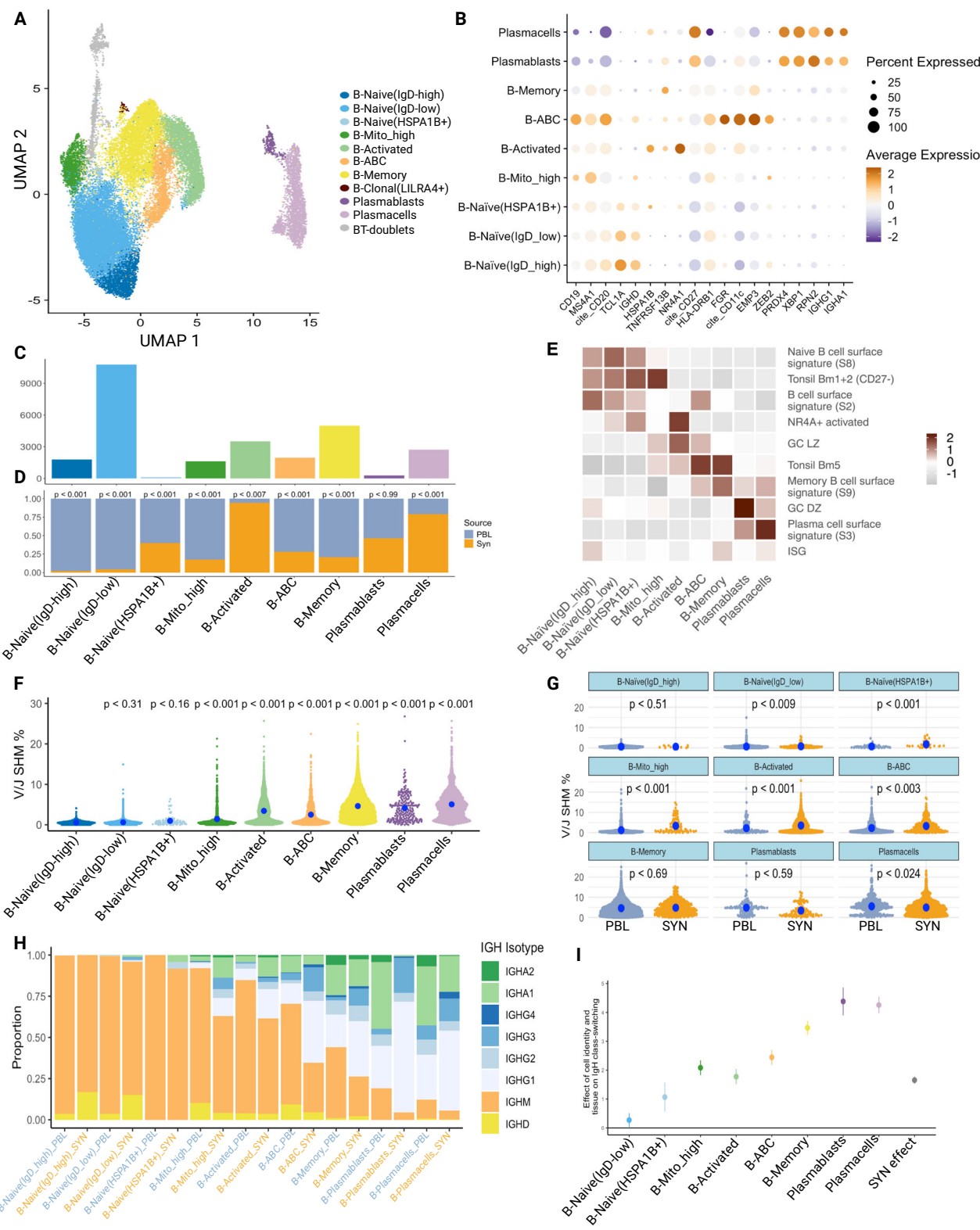

differentially expressed genes in the ABC cluster is *IFI30*, highlighting activation of interferon signaling as a potential driver of ABCs in situ (Supplementary Fig. 8C). The B-cell activated cluster was annotated based on high expression of *NR4A1*, related early response and germinal center light zone (GC LZ) genes (Fig. 5B, E and Supplementary Data 3). Notably, both ABCs and activated B cells showed evidence of class-switch recombination (Fig. 5H, I), consistent with previous reports[7,63]. Another small B cell population

was named clonal-*LILRA4*+ based on high BCR clonality and high *LILRA4* and G-protein signaling molecules. This small population only appeared in three PBL samples, with 87 out of 89 total cells belonging to one sample, and was omitted from downstream analyses. We used *XBP1* and *CD27* expression to identify plasmablast (*XPB1+CD27int*) and plasma cells (*XBP1+CD27hiIgA + IgG+*). Plasmablasts also mapped to a GC dark zone (GC DZ) signature (Fig. 5E), likely reflective of a high proliferation state (see Supplementary

**Fig. 5 | Accumulation of activated, somatically mutated B cells in the RA synovium. A** UMAP projection of B cell clustering. Unlabeled gray cells ("NA") were called B-T doublets as determined by high expression of *CD3E* and high doublet scores; these cells were omitted from downstream analysis. The population labeled B-Clonal(*LILRA4*+) was also omitted from downstream analysis due to having a small number of cells which were only found in two blood samples. **B** Dot plots of salient markers used in annotating clusters. **C** Bar plot of the number of cells included in each cluster. **D** Bar plot of the tissue distribution within each cluster. Significance determined by a two-sided mixed-effect model using MASC, shown in Supplementary Fig. 8D. **E** Heat map showing the scaled module score of select gene signatures. Heatmaps of selected genes for pathways are available in Supplementary Fig. 10A. **F** Plot quantifying differences in SHM rate between clusters (blood and synovial cells combined). *P* values were assessed through a two-sided linear

mixed-effect model with a random effect for donors. Reference population set to B-Naive(IgD-high). The blue dot represents a median. **G** Plot quantifying differences in SHM rate between tissues within each population. *P* values were assessed through a two-sided linear mixed-effect model with a random effect for donors. **H** Bar chart of productive immunoglobulin heavy chain (IgH) isotype usage for each cluster split by tissue after QC. **I** Plot of effect sizes (center of the dot) and 95% confidence intervals (error bar) for a mixed-effect logistic regression model which regresses a cell's class-switch state (IgG or IgA -> "class-switched", IgM or IgD -> "not class-switched") on its phenotype and tissue source. Random effects included for donors (*n* = 12 donors), B-Naive(IgD-high) set as the reference population and PBL set as the reference tissue source. PBL peripheral blood lymphocytes, SYN synovial tissue.

Fig. 9A for specific genes). Both populations exhibited high levels of class-switch recombination and SHM (Fig. 5F–I and Supplementary Data 3).

Next, we tested each population for enrichment in the blood versus synovium. All populations were found in both the blood and synovium and across multiple donors (Supplementary Fig. 8A). Activated B cells and plasma cells were significantly more abundant in the synovium. In contrast, Naive-IgD-low, Naive-IgD-high, B-memory, and Naive-MT-high clusters were significantly enriched in the blood (Fig. 5D and Supplementary Fig. 10D). We next sought to identify features that differentiate synovial from blood B cells by performing GSEA (Supplementary Fig. 10B). Many of the B cell populations in the synovium showed enrichment for GO pathways associated with cell activation and cytokine-mediated signaling (Supplementary Fig. 10B). The former is consistent with the overall activated state of B cells in the synovium. It is interesting that the most significant enrichment for cytokine-mediated signaling in any synovial B cell state is within the plasmablasts, suggesting that cytokine signaling is a critical mediator of plasmablast generation in the tissue (Supplementary Fig. 10B).

## Accumulation of somatic hypermutation and class switch in synovial B cells

A unique feature of this dataset is the ability to match the precise BCR sequence to the transcriptomically defined B cell states in both synovium and paired blood. Thus, we can determine which populations acquire SHM, the nature of those mutations, and the differences between tissue and blood. As expected, memory B cells and plasmablasts/plasma cells had significant levels of SHM. Of additional interest, activated B cells and ABCs also had higher levels of SHM compared to the Naive-IgD-high population as a reference (Fig. 5F).

We then compared the rate of SHM between blood and synovial cells for each population (Fig. 5G) Interestingly, three naive-like B cell populations—Naive-IgD-low, Naive-*HSPA1B*+, and Naive-MT-high— had significantly higher mutation rates in the synovium compared to blood, suggesting a spectrum of in situ naive B cell activation as we recently described[7]. Notably, ABCs and activated B cell populations also exhibited significantly higher mutation in the synovium, suggesting in situ activation and selection. The only population that had significantly higher mutation rates in the blood were plasma cells. Naive-IgD-high, memory, and plasmablasts did not show significance for tissue-specific mutation differences (Fig. 5G).

Additionally, we examined the amount of class-switching occurring within each cell population and between blood and synovial cells. Synovial B cell states exhibited significantly more class-switched BCRs (IgG or IgA+, IgD−) compared to their blood counterparts across all populations, except for naive-IgD-high B cells (Fig. 5H). All cell populations showed a statistically significant increase in the amount of class-switched BCRs compared to the least class-switched population: naive-IgD-high B cells (Fig. 5I). Of note, over 50% of the synovial ABCs showed evidence of class-switching consistent with the emerging concept that ABCs generated during normal immune response are a

subset of memory B cells[65,66]. It is also of interest that synovial ABCs, plasmablasts, and plasma cells have very distinct IgH isotype usage compared to their blood counterparts, with far fewer IgA and more IgG (Fig. 5H).

## Evidence of in situ antigen exposure and clonal expansion in synovial B cells

Using the BCRs recovered in our dataset, we identified groups of clonally related B cells by quantifying the similarity between their IgH CDR3 sequences. As a first approach to this analysis, we employed a stringent 96.5% sequence homology to identify clones. This was done in order to identify identical clones and thus establish developmental relationships between cell states. Cell populations that were more highly mutated produced larger B cell clones, as has been reported in analyses of blood B cell repertoires from healthy individuals (Fig. 6A, B)[67]. For example, plasmablast and plasma cells, which were recovered from nearly all samples (Supplementary Fig. 8A), had a higher proportion of large clones (20–100 cells) than other populations, consistent with antigen-driven clonal expansion (Fig. 6A, B). For plasmablasts/plasma cells, clonal expansion was more prominent in the blood than in the synovium (Fig. 6B). This suggests that plasma cells in the synovium are experiencing different selection pressures than those circulating in the periphery, also supported by the distinct IgH isotype expression (Fig. 5H). In the synovium, B cell populations other than plasma cells had mostly singletons or, within the activated B cells, ABCs, and MT-high, smaller clones (2–5 cells). Overall, the presence of both clonal expansion and higher SHM is consistent with stimulation and provision of T cell help within these more activated B cell populations. Further, consistent with an antigen-experienced B cell repertoire, activated B cells and ABCs had shorter CDR3 length and higher charge overall in both the synovium and blood (Supplementary Fig. 10A)[68]. In the blood, clones were also mostly singletons but with a proportion of small, medium, and even large clones detected in the activated, memory, MT-high, ABCs, and Naive-*HSPA1B*+ B cells (Fig. 6B).

To assess clonal relationships between cell types, co-occurrence of expanded clone members between cell types was reported for each clone that contained a member within two different cell types. Though the vast majority of clones were contained within a specific population, we did identify clones that were shared between populations. Figure 6C depicts the clonal sharing across populations within each compartment. Within the synovium, plasma cells and plasmablasts share a large number of clones (24 clones, Fig. 6C), strongly supporting a developmental relationship between newly generated plasmablasts and more mature plasma cells. Notably, we also observed shared clones between ABCs and activated B cells, as well as between both these B cell states and the plasma cells. A small number of clones are also shared between the MT-high B cell state and both activated B cells and ABCs (Fig. 6C, D). Overall, this data suggests a developmental relationship from Naive-MT-high (a naive B cell population already showing some signs of antigenic stimulation based on higher SHM rates compared to resting naive—Fig. 5F) alternatively down an ABC vs.

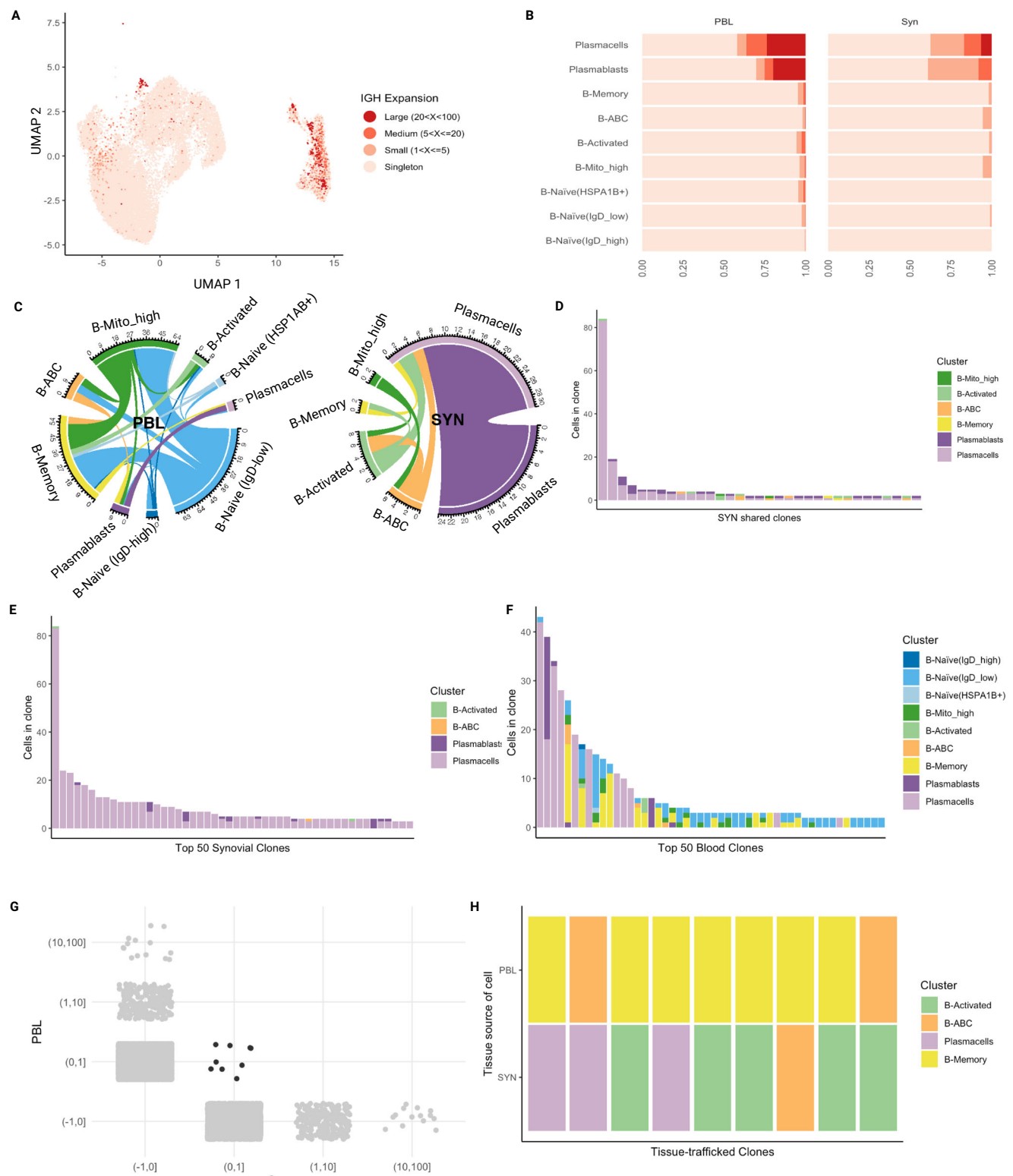

**Fig. 6 | Clonal expansion and clonal sharing between B cell states in the synovium. A** UMAP projection of IgH-defined clonal expansion in B cells. Cells without BCR are not included in this plot or other plots in this figure. **B** Bar chart showing the proportion of clone sizes for each cluster split by tissue. **C** Circos plot showing clonal sharing between cell populations split by tissue compartment. The color of the edge corresponds to the proposed originating population. **D** Bar plot showing the cell identity and clone size for each synovial clone shared by two or more B cell states. **E** Bar plot displaying the 50 largest clones in the synovium and their cell population composition. **F** Bar plot displaying the 50 largest clones in the blood and their cell population composition. **G** Plot displaying the distribution of clones across each tissue (gray) and clones shared between tissues (black). **H** Plot of cell population composition of the tissue-trafficked clones (black clones in **G**). PBL peripheral blood lymphocytes, SYN synovial tissue.

activated B cell pathway, as well as between these cell states and then downstream to plasma cells. It is also of interest that memory B cells share clones with activated B cells and plasma cells, consistent with memory B cells participating in synovial immune reactions (Fig. 6C, D). Within the peripheral blood, there is substantially more clonal sharing between multiple cell states (Fig. 6C). This is also evident when examining the cell population composition of the 50 most dominant clones in the blood compared to the synovium (Fig. 6E, F). As an example, the 5th most expanded blood clone was observed in the ABC, plasmablast, Naive-IgD-low, MT-high, and memory B cells (Fig. 6F). In contrast, the 50 most dominant synovial clones are heavily weighted toward plasma cells (Fig. 6E). Though clones were shared between the synovium and blood, these clones were not expanded (Fig. 6G). Tissue-trafficking clones were found within plasma cells, ABCs, and activated and memory populations, consistent with some trafficking of these B cells between the two compartments (Fig. 6H). Interestingly, clones shared between blood and tissue had distinct phenotypes (Fig. 6H). As an example, there are multiple cases of memory B cells in the blood becoming activated B cells or ABCs or plasma cells in the synovium, again strongly suggesting that memory B cells participate in synovial immune reactions.

In order to identify features that differentiate clonally expanded from unexpanded B cells in the synovium, we performed GSEA. We focused this analysis on plasmablasts and plasma cells, given the limited numbers of clones in other cell states. Clonally expanded plasmablasts showed increased expression of signatures of BCR signaling, cytokine-mediated signaling, and response to cytokines as compared to non-clonally expanded cells (Supplementary Fig. 11).

We repeated the analysis of clonally related B cells using a less stringent IgH CDR3 DNA sequence similarity of 80%[69]. The focus here was to generate and analyze clonal lineages in order to begin to map in situ immune reactions. Using this approach, more frequent small clones were detected in multiple cell states in both blood and synovium (compare Fig. 6B to Supplementary Fig. 12A, B). There were also more shared clones between cell states in the synovium, with sharing now detected from the naïve IgD-low and frequently across three or more populations (compare Fig. 6D to Supplementary Fig. 12C). As with 96.5% homology analysis, the most highly expanded synovial clones were in the plasmablast and plasma cell compartments (compare Fig. 6E to Supplementary Fig. 12D). With clonal lineages defined at 80% homology, more tissue-trafficking clones were identified (compare Fig. 6G, H to Supplementary Fig. 12E, F), again with examples of transitions of cell states between blood and tissue clones. Examples of B cell lineage trees highlight the developmental relationships from naïve IgD-low to ABC, activated, and plasma cells (Supplementary Fig. 12G, H).

### Identification of altered T cell-B cell communication patterns in synovium

Given the association of BCR signaling, B cell activation, and response to cytokines with the enriched and clonally expanded B cells in the synovial analysis, we next sought to systematically investigate potential T cell-B cell interactions. We constructed cell–cell communications networks using CellChat[70], with an initial focus on communication differences between the CD4 + T cell subpopulations and total B cells to assess the viability of the technique to first decipher well-characterized interactions. The proliferating, Tph, and Tfh/Tph subsets had the largest numbers of significant interactions identified with B cells (Fig. 7A). Detection of significant *CXCL13-CXCR5* interactions between Tph-B cell and Tfh/Tph-B cell pairs is consistent with prior reports and supports the performance of the analysis method[16]. A significant *IFNG-IFNGR* interaction was further detected between Tph-B but not Tfh/Tph-B pairs, consistent with the increased expression of *IFNG* in the Tph cluster compared to the Tfh/Tph cluster (Supplementary Fig. 13A, B). We then generated an inferred communication

network using CD4 + T subpopulations and B cell subpopulations. Examination of the cumulative incoming and outgoing interactions for each population identified the Tph population as having elevated signals for both directions, while the ABC population had the strongest outgoing signal of any cluster (Supplementary Fig. 13B). A pairwise analysis of significant interactions between these CD4 + T and B cell populations found the proliferating cluster to have the largest number of predicted interactions with all B cell subsets, though this may reflect a heterogeneous nature of proliferating cells. Aside from the pro-liferating cluster, elevated interactions between Tph cells and the ABC, plasmablast, memory, and clonal B cell populations were identified. The number of significant interactions for each of these pairs was higher than with any other CD4+ population, suggesting an elevated signaling potential of the Tph cluster (Fig. 7B).

Next, we leveraged the cross-tissue nature of our dataset to compare T cell-B cell signaling differences between synovial tissue and blood. We generated a separate communication network for each, both of which returned roughly similar numbers of significant inter-actions detected with a slight elevation in the synovium (Supplementary Fig. 13D). Within the synovial tissue, the TNF, CXCL chemokines, IL-2, and IFN type II signaling pathway families were elevated, together highlighting characteristics of an inflammatory and immune-activated state of the tissue. Among the signaling pathways underrepresented in the tissue compared to signaling in the blood were CCL, SELPLG, ICAM, and ITGB pathway families, which are often expressed in cells migrating in the blood (Fig. 7C). Finally, we sought to identify cell–cell interactions elevated in the synovial tissue between Tph or Tfh/Tph cells and either ABC, Memory, or Activated B cells. In all analyzed pairs, *BTLA-TNFRSF14* (HVEM) and *CXCL13-CXCR5* interactions were elevated in the synovium compared to the blood. In contrast, *ITGB2-ICAM2* and *LGALS9-CD44/45* interactions were higher in most blood pairs. Com-paring different interactions between Tph-B subsets and Tfh/Tph-B subsets, *LTA-TNFRSF14* (HVEM) was identified in all synovial Tph-B cell pairings but absent in Tfh/Tph-B cell pairings. In the synovium, *IFNG-IFNGR* was a significant interaction in both Tph/ABC and Tph/Activated B cell pairings, and *LTA-TNFRSF1B* (TNFR2) was specific to the Tph/ABC pairing only (Fig. 7D).

## Discussion

By leveraging paired single-cell RNA and TCR/BCR sequencing and paired synovium and blood samples, this work provides a detailed assessment of the relationship between the immune repertoire and cell state composition, gene expression, blood and synovial lympho-cyte trafficking, and cell–cell interactions in RA synovium. These data provide insight into the developmental relationships between specific synovial lymphocyte populations, for example, demonstrating clonal links connecting ABC and activated B cells with plasma cells, while separating clonally distinct Tph/Tfh vs *CCL5* + CD4 T cells and *GZMK+* vs *GZMB* + CD8 + T cells (Fig. 8).

Within the CD4 T cell compartment, the Tph cells express among the highest effector and activation signatures compared to other subsets and are significantly enriched in synovium compared to the blood, consistent with prior observations[16,71]. Here, we leverage the paired repertoire information of this population, identifying it as one of the most clonally expanded CD4 T cell populations and clonally related to both proliferating cells and Tfh cells. We acknowledge that a potential limitation of our approach is variability in the number of synovial T cells (and B cells) isolated and analyzed from different samples, which can affect the clone sizes detected. Nonetheless, we observed that Tph cells from multiple synovial tissue samples were clonally expanded and related to CXCR5+ Tfh cells, consistent with recent work tracking clonal relationships of Tph cells in synovial fluid[29]. Analyses of synovial tissue performed here suggest that Tfh cells, as contained within the Tfh/Tph cluster, may be more abundant in synovial tissue than in synovial fluid, in which PD-1hi CD4 + T cells

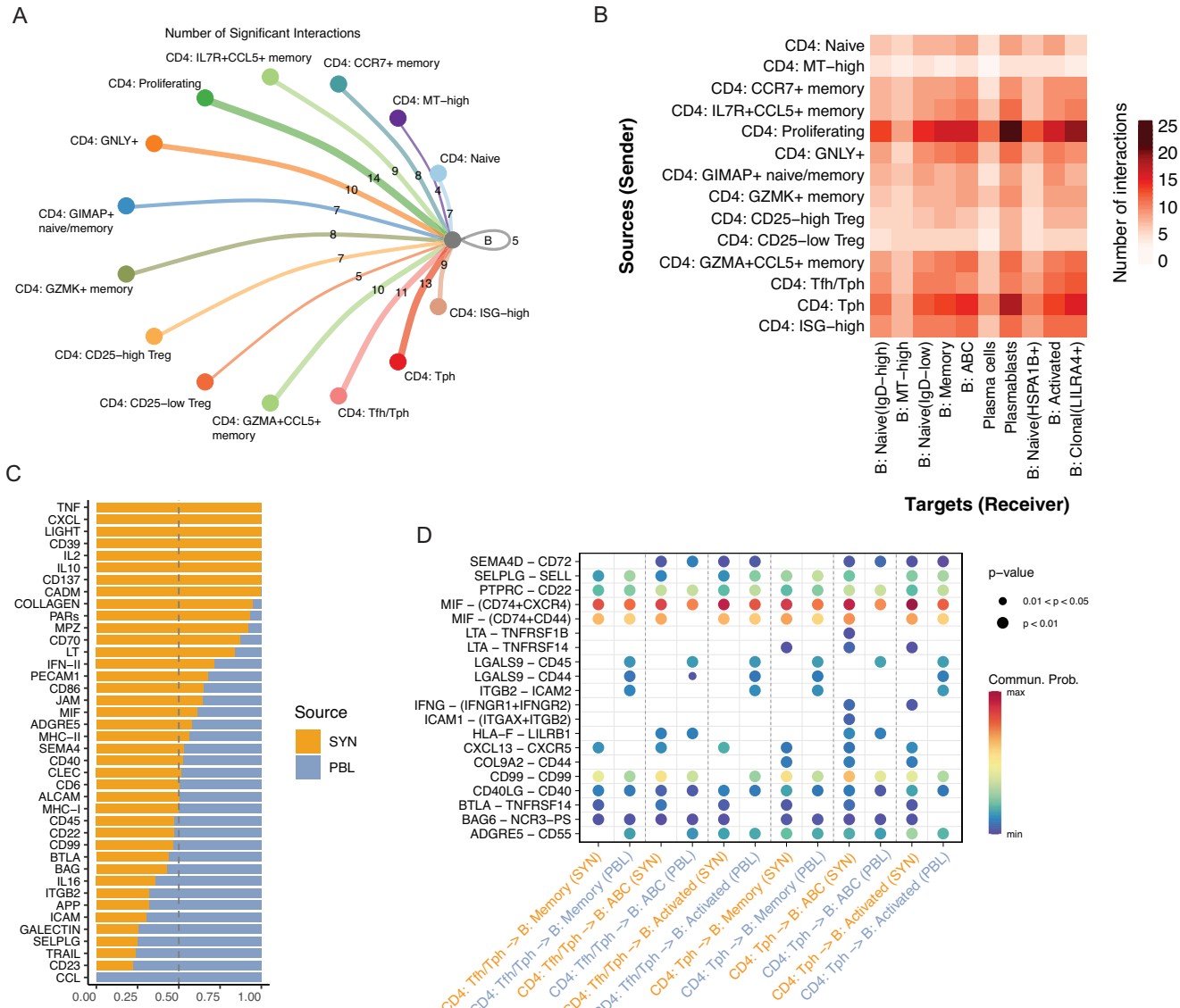

**Fig. 7 | Identification of altered T-B cell communication pathways. A** Circle plot of the number of significant cell–cell interactions identified between CD4 + T cell subsets and B cells. **B** Heat map of the number of significant cell–cell interactions identified between CD4 + T cell subsets as senders and B cell subsets as receivers. **C** Bar plot of the proportion of a signaling pathway's detection in synovial tissue or blood. **D** Dot plot of significant interactions detected between Tph or Tfh/Tph cells and ABC, Memory, or Activated B cells, split by tissue, as assessed by a permutation test ($n$ = 100 permutations).

are predominantly CXCR5- Tph cells. Further, our identification of some shared clones between Tph in synovium and in blood, even with small numbers of total T cells analyzed, supports the notion that a portion of Tph cells in blood have both transcriptomic and clonal relationships to Tph cells in synovium.

The gene expression patterns in clonally expanded Tph cells, combined with predicted cell–cell interactions, suggest additional roles for Tph cells, where upon T cell activation and clonal expansion, they upregulate factors related to effector function and cytotoxicity, such as *GZMB*, *IFNG*, and *CCL5*. This result may suggest a direct contribution by this population in promoting tissue inflammation or injury through cytotoxic activity in addition to its well-characterized B cell helper function (Fig. 8). Additionally, the IFNG production by activated and clonally expanded Tph cells could drive ABC production, while CCL5 production could attract additional effector T cells and myeloid cells.

While analyses of CD8 T cells have largely focused on the expression of cytotoxic features such as granzyme B, recent

observations have highlighted a prominent CD8 T cell population in RA synovium with distinct expression of granzyme K[22]. Our analyses further underscore a key distinction between *GZMK+* and *GZMB* + CD8 + T cells in RA synovium and blood. Whereas in the blood *GZMB*-expressing cytotoxic cells formulate the largest clones, CD8+ cells that express *GZMK* comprise the majority of the most expanded cells in the tissue. A striking finding through the current and prior analyses was the near-absence of clonal overlap between *GZMK+* cells and *GZMB+* cells across tissues, suggesting that *GZMK+* cells do not arrive at the synovium as *GZMB*-expressing cytotoxic cells and may instead receive antigenic stimulation locally that drives clonal expansion and functions such as cytokine production[22].

The presence of T cell cross-reactivity between virus and self has previously been associated with the capacity for driving auto-immunity, including in RA, yet the potential for virus-reactive T cells to contribute to synovial inflammation has remained uncertain[72–75]. Our analyses identify multiple synovial T cell clones that match those previously identified to be viral-reactive, a result consistent with the

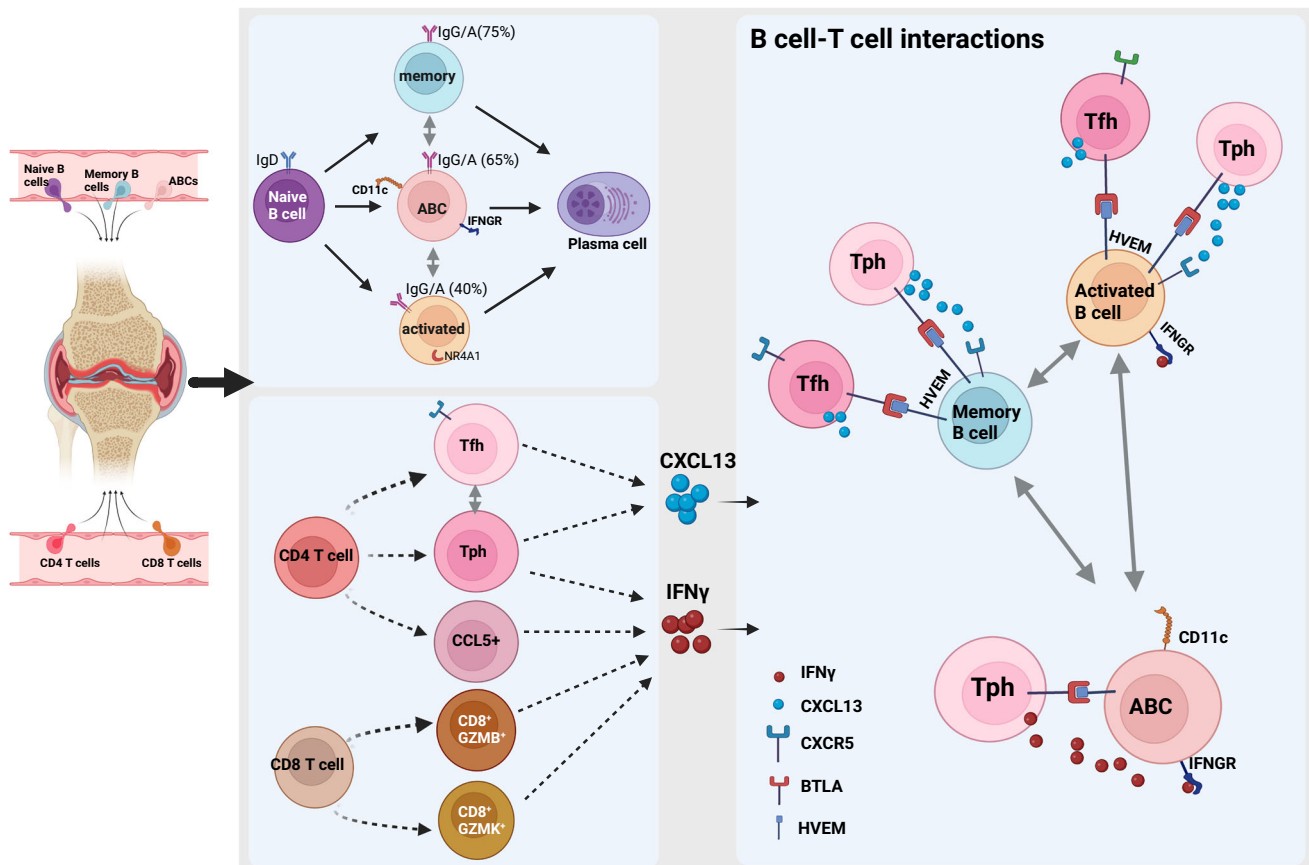

**Fig. 8 | Clonal associations of lymphocyte subsets and functional states revealed by single-cell antigen receptor profiling of T and B cells in RA synovium.** Multiple B cell and T cell subsets are enriched in the RA synovium including autoimmune associated B cells (ABC) and Tph cells. Clonal analyses of B cell repertoire indicate relationships between memory, activated, and ABC B cells with plasma cells in the synovium. Additionally, memory B cells and ABCs have a high percentage of class-switched clones (75 and 65%, respectively) (compared to 0% for naive, 40% for activated B cells, and >95% for plasma cells). Specific CD4 and CD8 T cell subsets are enriched in RA synovium. Synovial Tfh cells and Tph cells show significant clonal relationships. Tph cells uniquely produce both IFNγ and CXCL13. These cytokines are important for the interactions with various B cell states, including ABC, activated, and memory B cells. Additionally, receptor-ligand analysis indicated that B cells and T cells in the synovium may interact through BTLA-HVEM binding, promoting synovial immune reactions. Figure 8, created with BioRender.com, released under a Creative Commons Attribution-NonCommercial-NoDerivs 4.0 International license.

recent demonstration of viral reactivity among TCRs from RA synovial CD4 T cells[33]. Our paired RNA-seq/TCR analysis enabled interrogation of the phenotypes of potentially viral-reactive CD8 T cells, yet no broad-scale difference in the cluster composition or clonal characteristics of these cells was apparent. Because our methodology relied on utilizing viral-specific clones that had previously been identified on defined HLA alleles, there are likely a number of "false-negative" clones in the dataset that could be viral-reactive yet did not have a match in the examined databases. Future work to better define the viral-reactive capacity of T cells within the joint may rely on isolating viral-specific cells using viral peptides bound to tetramers and evaluating them for cross-reactivity to synovial antigens. Still, this work shows the presence of likely cross-reactive T cells within the synovium, yet with no enrichment for specific activated or effector phenotypes.

Populations of innate T cells are thought to contribute to RA[76,77]. Here, we define the subsets of innate T cells present in the synovium, including populations of γδ T and MAIT cells, with selective enrichment of Vdelta1 γδ T cells in the synovium compared to blood, consistent with their enrichment in other tissues including gut and skin[78,79]. Leveraging paired TCR information, this is the first single-cell RNA sequencing study in RA to confirm the presence of MAIT cells by analyzing VDJ gene rearrangements. Our detection of shared MAIT cell clones in synovium and blood suggests that these cells may traffic into and out of synovium. Synovial MAIT cells showed an increased

activation signature compared to those in blood, as did γδ T cells, suggesting an active role for these cells in inflammatory arthritis.

One of the striking findings of our study is the enrichment of activated B cell populations in the synovium with evidence of clonal expansion and clonal sharing between different B cell states. We achieved unprecedented resolution of discrete B cell states, and in contrast to a previous study[36], our data demonstrated clonal sharing between multiple B cell states extending beyond memory B cell and PC pools. We further extend the observation from previous studies[34,36] that synovial plasma cells are generated from locally activated B cells, including activated B cells, ABCs, and memory B cells. The finding that the ABCs are a precursor of plasma cells in the synovium is consistent with a recent report[80]. Further, our work highlights the likely important signals promoting synovial B cell activation and selection, including antigen (reflected in upregulation of BCR signaling), cytokines (most notably *IFNG*), and direct cell–cell interactions, mainly involving Tph/Tfh cells.

Our study highlighted a spectrum of B cell activation unique to the synovium. We identified multiple naïve-like B cell states, with surprising evidence of antigen encounter/activation based on higher mutation rates. The majority of the synovial B cells in our study are non-naive, dominated by a B-activated cluster expressing *NR4A1*[7], ABCs, and plasmablasts/plasma cells. The higher mutation rates in these synovial enriched B cell subsets compared to their blood counterparts suggests that these cells are under different selection

pressures. Although it is possible that more activated, somatically mutated B cells preferentially home to the synovium, we favor the hypothesis that activation and SHM occurs in situ, also supported by synovial clonal expansion and clonal sharing across cell states. Of note, the synovial B cell SHM is lower than germinal center B cells in the tonsil and similar to B cell mutation rates reported in other inflamed tissues[81–83]. We have hypothesized that activated B cells are involved in initiating ectopic lymphoid structures (ELS), as evidenced by their higher expression of ELS-inducing cytokines like LT and IL6[7]. The data presented here further highlight that *NR4A1*+ activated B cells may go down an extrafollicular pathway as they share clones with ABCs and have a lower mutation rate compared to tonsil germinal center B cells. The availability of paired analysis of T and B cells from the same synovial samples allowed us to directly link clonally expanded T cell populations with synovial tissue expanded B cells for the first time. The most striking predicted interactions were identified between proliferating T cells and Tph with activated B cells, ABCs, and plasmablasts. Notably, Tph and ABCs had among the largest numbers of incoming and outgoing interactions, suggesting an elevated signaling potential for these populations. Signaling pathways enriched in the synovium and identified in the Tph-ABC interaction included cytokines (e.g. *IFNG-* produced by Tph and *IFNGR* on ABCs, *LTA-TNFRSF14* (HVEM), and *BTLA-TNFRSF14* (HVEM) interactions) and chemokines (see schematic in Fig. 8). Notably, GSEA also identified cytokine-mediated signaling as prominent in clonally expanded synovial B cells, including within the ABC population. This subset of B cells has been previously reported to be expanded in autoimmune disease[63] with accumulation in inflamed tissue[24,26], but the signals that promote ABCs in the synovium remain unclear. Our data strongly point to Tph cells as a key driver. This is in accord with a recent report in juvenile idiopathic arthritis demonstrating that clonally expanded *IL21* and *IFNG* co-expressing Tph promote CD11c+ double negative B cell differentiation[84].

Together, these findings across T and B cells highlight the altered cell state composition and clonal characteristics that may work together to maintain inflammation in RA. Our study utilized a cross-sectional cohort unified in high disease activity, but otherwise heterogeneous across treatment history, disease duration, and cell-type abundance phenotype (CTAP). Further studies with a larger cohort of patients will be necessary to connect clonal characteristics such as those identified here with patient stratifications, which may serve to increase our knowledge of the inherent cellular and molecular heterogeneity of the disease. It will also be of interest to define the specificity of the B and T cells that are clonally expanded, including their reactivity to citrullinated antigens. Citrullinated peptides may be the main antigenic drivers in the synovium as has been suggested by other studies[6,85]. As the understanding of pathogenic roles of B and T cell subsets in RA continues to evolve, this dataset will be a useful resource to generate or test insights related to the antigen receptor repertoires of synovial lymphocytes. The work further highlights specific lymphocyte populations, including Tph cells, ABC, and activated B cells, that show both transcriptomic signatures of antigen activation and clonal expansion, marking these cell populations as promising therapeutic targets that might be selectively targeted to blunt the pathologic adaptive immune response in RA.

## Methods

### Sample processing

For this study, patients were recruited and consented through the Accelerating Medicines Partnership (AMP) Network for RA and SLE[26]. Samples were collected from 15 clinical sites who are part of the AMP Network. The study was performed with informed consent in accordance with protocols approved by the Institutional Review Board at Stanford University. Written informed consent was obtained from all participants. Synovial tissue samples and matched peripheral blood

mononuclear cells were cryopreserved after collection as described[86]. Stored synovial tissue samples were then thawed and disaggregated into single-cell suspensions by mincing and digesting with 100 μg/mL LiberaseTL (Roche) and 100 μg/mL DNaseI (Roche) in RPMI (Life Technologies) for 15 min, with occasional inversion during disaggregation. Disaggregated cells were passed through a 70 μm cell strainer and washed prior to antibody staining with anti-CD235a antibodies (clone 11E4B-7-6 (KC16), Beckman Coulter) and Fixable Viability Dye eFluor 780 (eBioscience/Thermo Fisher). Live non-erythrocyte cells (viability dye- CD235-) were collected by fluorescence-activated cell sorting (BD FACSAria Fusion) and were initially cryopreserved in Cryostor CS10 (Sigma-Aldrich). The disaggregated synovial tissue cells and matched cryopreserved peripheral blood mononuclear cells were then thawed in batches, and both T and B cells were collected by fluorescence-activated cell sorting (BD FACSAria II) (Supplementary Fig. 1B). A description of the antibodies used in this study can be found in Supplementary Data 4. A maximum of 16,000 of B and T cells combined were sorted from each sample. We sorted 8000 B cells and 8000 T cells for the majority of blood samples. For synovial samples, the number of B cells and T cells sorted varied depending on cell abundance (B cell: 107-7472 cells, T cell: 2616-13636 cells) (Supplementary Fig. 1C, D). B cells and T cells from each sample were pooled prior to loading on a Chromium NextGEM Chip G (10X Genomics).

### Single-cell library preparation and sequencing

Cells collected from cell sorting were encapsulated into oil droplets using a Chromium NextGEM Chip G (10X Genomics). Following reverse transcription and cDNA amplification, 5' gene expression, immune repertoire, and feature barcode libraries were constructed following manufacturer protocols (v1.1). The libraries were finally pooled for sequencing on an Illumina Novaseq 6000 using an S4 flow cell. Gene expression libraries were sequenced to obtain a read depth of 100,000 reads per cell, feature barcode libraries were sequenced at 5000 reads per cell, and immune repertoire libraries were sequenced at 5000 reads per cell. FASTQ file demultiplexing for gene expression libraries was performed using the *mkfastq* function in CellRanger (10X Genomics, v4.0). Following this, alignment to a reference genome (GRCh38) and counting was completed using the *count* function to generate expression matrices for each sample. Immune repertoire FASTQ files were separately demultiplexed, and the *vdj* function was used to perform sequence assembly and clonotype calling for TCR and BCR sequences in each sample.

### Initial quality control

Gene count matrices were imported in R for downstream analysis. Quality control was first performed jointly on all cells collected and sequenced in this experiment. Several metrics were explored to assess the quality of each cell. First, low-quality cells were distinguished from high-quality cells in each tissue compartment. Cells were kept for downstream analysis if they had at least 1000 mapped reads in either the blood or the synovium. After initial filtering, we generated several metrics to identify doublets using software packages scDblFinder[87] and scds[88], as well as marking cells that coexpressed at least 1 TCR and 1 BCR. At this point, a single synovial sample (AMP ID# 300_0415) was discarded due to only 12 cells passing these initial QC thresholds. Log-normalization was then applied to the gene expression counts for the remaining cells. Final QC thresholding was performed on the log-normalized counts, keeping cells with greater than 500 genes detected and less than 20 percent of detected reads coming from mitochondrial genes. Additional quality control was performed for T and B cells separately in downstream analysis.

### Broad cell type identification

After initial QC, unsupervised clustering was performed on the remaining cells to identify major cell types present in the data.

Principal components were first generated in order to reduce the dimensionality of the feature space before clustering. Using Seurat's clustering functionalities on the first 30 principal components, a 20 nearest-neighbors network graph was computed. Then, we performed Louvain clustering with a resolution parameter of 0.3, and visualized the cells in 2D space using uniform manifold approximation and projection (UMAP). Differentially expressed genes between clusters were identified (Student's *T*-test) by including only genes exhibiting a greater-than 0.25 log-fold difference between clusters. In order to annotate each cluster with a biologically meaningful name, genes with the highest log-fold changes were considered, as well as marker genes that are cell-type specific.

### T cell subclustering

Cells broadly labeled as "T cells" and "Proliferating" from the combined object were subset, and Harmony[89] was then used to perform batch correction at the level of the patient and tissue using theta = 2 and max.iter.cluster = 20. Using the top 50 Harmony embeddings, Louvain clustering was performed, which was then visualized in UMAP space. Nearest neighbors were identified using the Harmony embeddings, and clustering was iteratively performed with a resolution of 1.4 finally selected. Broad T cell markers (e.g., CD4, CD8, TRDC) were used along with differential gene expression (Wilcoxon rank-sum test) to identify the T cell lineages for each of these initial clusters. At this point, a small number of remaining contaminating cells (such as B cells from the Proliferating cluster), were removed. Based on gene expression, clusters of CD4, CD8, and innate T cells were separated and individually clustered using a similar strategy with slightly different parameters for each subset (CD4: 40 Harmony embeddings, 0.5 cluster resolution; CD8: 40 Harmony embeddings, 0.4 cluster resolution; innate T: 10 Harmony embeddings, 0.4 cluster resolution).

### B cell subclustering

Cells labeled as "B cells" from the broad clustering, we further characterized B cell subpopulations. Before reclustering the B cells, we discard cells marked as doublets according to scDblFinder and B cells which simultaneously coexpressed at least 1 BCR and at least 1 TCR. On the remaining cells, Seurat's default normalization and scaling was performed and principal components were generated. Harmony was used to perform batch correction at the patient level using theta = 2 and max.iter.cluster = 20. From here, the 20 nearest-neighbors network graph was generated using the first 30 harmonized principal components. We then applied Louvain clustering using a resolution parameter of 0.5 to identify clusters of similar cells to visualize in the UMAP space. Differential gene expression was then performed (Student's *t*-test) to provide markers for cluster annotation. Before selecting the final set of input parameters, results were explored at multiple resolutions, variable number of included principal components, and using both harmonized and non-harmonized principal components.

### Gene signature analysis

Gene signatures used in this study were obtained from the sources listed in Supplementary Data 3. The *AddModuleScore* function in Seurat was used to assign a value for each cell for each signature, corresponding to the average expression of the signature subtracted by the aggregate expression of randomly-selected control genes.

### Dataset reference mapping

A reference object was built using Symphony[90] with cells from ref. 26 that correspond to the populations being assayed (CD4 T, CD8 T, innate T, or B cells), integrating at the sample level and using the first 20 PCs. Data from the current study was then projected onto this reference using the mapQuery and knnPredict functions, to generate confidence scores for each reference cluster's mapping, which we

further visualized using pheatmap. The most-likely identities for each cluster were then cross-referenced with DEG lists from our dataset to aid in generating final cluster identities.

### Single-cell TCR receptor profiling

For each sample, the filtered_contig_annotations.csv file output from cellranger was used to identify TCR sequences obtained for each cell barcode using scRepertoire[91]. Using the filterMulti argument, only the top 2 expressed chains were retained when a cell barcode was associated with more than two chains (e.g., ααβ or αββ). This step also worked to remove TCR information when only 1 chain was available for a specific cell barcode (e.g., α only or β only). These TCRs and their cell barcodes were then matched with corresponding cell barcodes obtained from the sample's RNA library. Combined, 45,096 cells in the initial T cell subset had available TCR information, including 22,634 cells obtained from synovial tissue and 22,462 cells obtained from blood. Clones were further characterized into discrete groups for their extent of clonal expansion for downstream analysis.

### Single-cell BCR receptor profiling

Analogous to the TCR profiling, output from CellRanger was used to identify BCR sequences for each cell barcode. For downstream analysis, we included BCR that were considered high-confidence, full-length, length of CDR3 amino acid at least 5, and length of CDR3 DNA at least 15. We also excluded BCR that were associated with more than two heavy chains or more than two light chains. BCR passing QC were then matched with cell barcodes from the sample's single-cell RNA library. Combined, 38,482 cells in the B cell subset had associated BCR information, of which, 12,357 cells were from the synovial tissue and 26,125 were from blood. We assigned each BCR sequence to its closest sequence in the IMGT database using the Change-O tool[92]. The degree of somatic hypermutation in each B cell was quantified by determining the number of V and J substitutions in each B cell's IgH CDR3 sequence when compared to its closest IMGT sequence. Clones were characterized in B cells according to the similarity of their CDR3 DNA regions. A similarity threshold of 96.5%, CD-HIT[93] was used to define discrete clonal groups for downstream analysis. Clonal analysis was also repeated using a similarity threshold of 80%.

### Clonal lineage tree analysis

In exploring the evolutionary relationships among BCR sequences within B cell clonal lineages, we utilized alignment and identity scores for both variable (V) and junctional (J) gene segments, along with somatic hypermutation (shm) levels within these gene segments. Additionally, we calculated the combined V and J somatic hypermutation rate (V-J shm rate), obtained by dividing the sum of V-shm and J-shm by the sequence length excluding the CDR3 region, all in comparison to the germline. A distance matrix was computed using the Euclidean method. Subsequently, we applied the ward.D method for hierarchical clustering to construct the dendrogram plot.

### Mixed-effect modeling

A number of mixed-effect models were fit in our analysis, which all generally took a similar form. To adjust for patient-specific effects in our data, mixed-effect models were fit using sample_ID as a scalar random effect and fixed effects for other covariates of interest. Lme4[94] was used to obtain point estimates for all mixed-effect models, with 95% confidence intervals.

### Gene set enrichment analysis

Gene set enrichment analysis was performed using fast gene set enrichment analysis (FGSEA)[95]. FGSEA calculates an enrichment score for each gene set, given a ranked vector of gene-level statistics. A null distribution of the enrichment score is estimated through random sampling of gene sets. *P* values are estimated as the number of random

gene sets with more extreme enrichment scores than the gene set of interest, divided by the number of random gene sets generated. Multiple testing correction is then performed to get adjusted $P$ values. We first performed differential expression for our comparisons of interest within each population (SYN vs PBL and clonal vs non-clonal) to obtain a ranked vector of gene-level statistics. We then sampled 1000 gene random sets to estimate the null distribution of each gene set's enrichment score and calculate adjusted $p$ values.

## Cell–cell interactions

Cell–cell interaction inference was performed using CellChat[70], which uses a manually curated database of ligand–receptor interactions gathered from KEGG signaling pathways and published literature. Interaction networks are constructed by identifying differentially expressed genes related to these interactions, computing the average expression of each ligand–receptor pair across cell cluster pairings, and finally calculating a communication probability value based on permutation testing. For finding communication probabilities, we used the tool's Tukey triMean method, which performs a weighted average of the median and upper and lower quartiles, and further used 100 bootstraps (nboot = 100) to calculate $p$ values.

## HLA imputation

To obtain estimated two-field HLA alleles in each donor, we performed HLA imputation from SNP genotype data. We genotyped donors from this study by using the Illumina multi-ethnic genotyping array. We performed quality control of genotype by sample call rate >0.99, variant call rate >0.99, minor allele frequency >0.01, and $P_{HWE} > 1.0 \times 10^{-6}$. We extracted the extended MHC region (28–34 Mb on chromosome 6) and performed haplotype phasing with SHAPEIT2 software[96](Delaneau, Marchini, and Zagury 2011). We then performed HLA imputation by using a multi-ancestry HLA reference panel version 2[97] and minimac3 software[98]. From the imputed dosage of two-field HLA alleles of each HLA gene in each donor, we defined the most likely set of two two-field alleles (Supplementary Data 5).

## Virus reactivity analysis

Publically-available datasets of TCRs were obtained from VDJdb[50] and McPAS[49], and were subsequently filtered to retain only TCRs associated with Epstein-Barr virus (EBV), cytomegalovirus (CMV), and influenza (FLU). As the large majority of data uploaded to these databases are from bulk TCR sequencing with only the β chain information available, we subsequently focused only on this chain in our single-cell TCR data. For exact matching, a "match" was considered when the CDR3 region, as well as the MHC allele, were the same. For GLIPH2 analysis, the same lists of EBV, CMV, and FLU reactive cells were input alongside patient CD8 TCRs as input. Results were filtered to retain GLIPH groups with previously identified TCRs presented on an MHC that matched an HLA allele from the patient. GLIPH group results were also filtered for stringency, with a Fisher score <0.01.

## Reporting summary

Further information on research design is available in the Nature Portfolio Reporting Summary linked to this article.

## Data availability

Single-cell RNA and TCR/BCR sequencing data generated in this study are available via the ARK Portal (https://doi.org/10.7303/syn47217489.1) The data were available under controlled access due to data privacy laws. To access the data, users need to complete and submit a signed Data Use Certificate (DUC) to the ARK Portal at https://arkportal.synapse.org/Data%20Access. Additional data are provided in the Supplementary Information and as Source Data files. Source data are provided with this paper.

## Code availability

The source code to reproduce analyses used in this study is available at https://github.com/dunlapg/amp2repertoire/tree/main.

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

## Acknowledgements

We thank the participants who provided synovial tissue and blood samples for this study. We also acknowledge the Flow Cytometry Core and Genomics Research Center at the University of Rochester for performing the flow sorting, scRNA library preparation, sequencing, and primary data analysis for this study. This work was supported by the Accelerating Medicines Partnership Program: Rheumatoid Arthritis and Systemic Lupus Erythematosus (AMP RA/SLE) Network. The AMP Program is a public-private partnership that includes AbbVie, the Arthritis Foundation, Bristol-Myers Squibb Company, the Foundation for the National Institutes of Health, GlaxoSmithKline, Janssen Research and Development, the Lupus Foundation of America, the Lupus Research Alliance, Merck, the National Institute of Allergy and Infectious Diseases, the National Institute of Arthritis and Musculoskeletal and Skin Diseases, Pfizer, the Rheumatology Research Foundation, Sanofi, and Takeda Pharmaceuticals. Funding for AMP RA/SLE work was provided through grants from the National Institutes of Health (UH2-AR067676, UH2-AR067677, UH2-AR067679, UH2-AR067681, UH2-AR067685, UH2-AR067688, UH2-AR067689, UH2-AR067690, UH2-AR067691, UH2-AR067694, and UM2-AR067678). F.Z. is supported by the PhRMA Foundation Faculty Starter Grant for Translational Medicine. K.W. is supported by a Burroughs Wellcome Fund Career Awards for Medical Scientists, a Doris Duke Charitable Foundation Clinical Scientist Development Award, and a Rheumatology Research Foundation Innovative Research Award. D.A.R. is supported by a Burroughs Wellcome Fund Career Awards for Medical Scientists.

## Author contributions

G.S.F., D.L.B., D.T., and C.P. recruited patients and obtained synovial tissues. L.W.M., S.M.G., V.M.H., A.F., V.P.B., and J.H.A. contributed to the procurement of samples and overall design of the AMP study. N.M., A.H.J., K.W., A.N., L.T.D., S.R., and M.B.B. contributed to the design and implementation of the tissue disaggregation, cell sorting, and single-cell sequencing pipeline. G.S.D., A.W., R.W., F.Z., A.H.J., S.S., J.T., and A.M. conducted the computational analysis. A.M., D.A.R., and J.H.A. supervised the study. G.S.D., A.W., N.M., J.C.E., D.A.R., and J.H.A. wrote and edited the initial manuscript draft. AMP RA/SLE Network members contributed to this study by managing patient recruitment, curating patient clinical data, obtaining and processing synovial tissue samples, managing biorepositories, providing website support, and/or providing input on data analysis and interpretation. All authors participated in revising the final manuscript.

## Competing interests

A.H.J. receives research support from Amgen unrelated to the submitted work. K.Wei received a sponsored-research agreement from Gilead Sciences, 10X Genomics, and served as a consultant for Mestag Therapeutics and Santa Ana Bio. S.M.G. reports research support from Novartis and is a consultant for UCB unrelated to this work. G.S.F reports receiving grant support from Eli Lilly. V.M.H. is a co-founder of Q32 Bio and previously received research support from Janssen and was a consultant for Celgene and BMS, outside the submitted work. A.F. reports personal fees from Abbvie, Roche, and Janssen and grant support from Roche, UCB, Nascient, Mestag, GlaxoSmithKline and Janssen unrelated to this work. M.B.B. is a founder of Mestag Therapeutics and a consultant for GlaxoSmithKline, 4FO Ventures, and Scailyte AG. S.R is a founder of Mestag Therapeutics, a scientific adviser for Janssen and Pfizer, and a consultant for Gilead and Rheos Medicines, D.A.R. reports sponsored research from Janssen, Merck, and Bristol-Myers Squibb unrelated to the current work, and reports personal fees from Pfizer, Janssen, Merck, Scipher Medicine, GlaxoSmithKline, and Bristol-Myers Squibb and is co-inventor on a patent using T peripheral helper cells as a biomarker of autoimmune diseases. A.M.M. is a promotional speaker for Abbie and Pfizer, have developed educational content and speak for the National Psoriasis Foundation, and served as a consultant for CVS Caremark which none of this is relevant to this study. The remaining authors declare no competing interests.

## Additional information

[1]Division of Rheumatology, Inflammation, and Immunity, Department of Medicine, Brigham and Women's Hospital and Harvard Medical School, Boston, MA, USA. [2]Department of Biostatistics and Computational Biology, University of Rochester School of Medicine and Dentistry, Rochester, NY, USA. [3]Division of Allergy, Immunology and Rheumatology, University of Rochester Medical Center, Rochester, NY, USA. [4]Department of Microbiology and Immunology, University of Rochester School of Medicine and Dentistry, Rochester, NY, USA. [5]Center for Data Sciences, Brigham and Women's Hospital, Boston, MA, USA. [6]Division of Genetics, Department of Medicine, Brigham and Women's Hospital, Boston, MA, USA. [7]Department of Biomedical Informatics, Harvard Medical School, Boston, MA, USA. [8]Broad Institute of MIT and Harvard, Cambridge, MA, USA. [9]Division of Rheumatology and the Center for Health Artificial Intelligence, University of Colorado School of Medicine, Aurora, CO, USA. [10]Hospital for Special Surgery, New York, NY, USA. [11]Weill Cornell Medicine, New York, NY, USA. [12]Division of Rheumatology, Allergy and Immunology, University of California, San Diego;, La Jolla, CA, USA. [13]Division of Rheumatology, University of Colorado School of Medicine, Aurora, CO, USA. [14]Division of Rheumatology and Clinical Immunology, University of Pittsburgh School of Medicine, Pittsburgh, PA, USA. [15]Centre for Experimental Medicine & Rheumatology, EULAR Centre of Excellence, William Harvey Research Institute, Queen Mary University of London, London, UK. [16]Barts Health NHS Trust, Barts Biomedical Research Centre (BRC), National Institute for Health and Care Research (NIHR), London, UK. [17]Department of Biomedical Sciences, Humanitas University and Humanitas Research Hospital, Milan, Italy. [18]Rheumatology Research Group, Institute for Inflammation and Ageing, University of Birmingham, NIHR Birmingham Biomedical Research Center and Clinical Research Facility, University of Birmingham, Queen Elizabeth Hospital, Birmingham, UK. [19]Birmingham Tissue Analytics, Institute of Translational Medicine, University of Birmingham, Birmingham, UK. [20]NIHR Birmingham Biomedical Research Center and Clinical Research Facility, University of Birmingham, Queen Elizabeth Hospital, Birmingham, UK. [39]These authors contributed equally: Garrett Dunlap, Aaron Wagner, Nida Meednu. [40]These authors jointly supervised this work: Andrew McDavid, Deepak A. Rao, Jennifer H. Anolik. ✉e-mail: darao@bwh.harvard.edu; jennifer_anolik@urmc.rochester.edu

## Accelerating Medicines Partnership Program: Rheumatoid Arthritis and Systemic Lupus Erythematosus (AMP RA/SLE) Network

Jennifer Albrecht[3], William Apruzzese[21], Jennifer L. Barnas[3], Joan M. Bathon[22], Ami Ben-Artzi[23], Brendan F. Boyce[24], S. Louis Bridges Jr[10,11], Debbie Campbell[3], Hayley L. Carr[18], Arnold Ceponis[12], Adam Chicoine[1], Andrew Cordle[25], Michelle Curtis[1,5,6,7,8], Kevin D. Deane[13], Edward DiCarlo[26], Patrick Dunn[27,28], Lindsy Forbess[23], Laura Geraldino-Pardilla[22], Ellen M. Gravallese[1], Peter K. Gregersen[29], Joel M. Guthridge[30], Diane Horowitz[29], Laura B. Hughes[31], Kazuyoshi Ishigaki[1,5,6,7,8,32], Lionel B. Ivashkiv[10,11], Judith A. James[30], Joyce B. Kang[1,5,6,7,8], Gregory Keras[1], Ilya Korsunsky[1,5,6,7,8], Amit Lakhanpal[10,11], James A. Lederer[33], Yuhong Li[1], Zhihan J. Li[1], Katherine P. Liao[1,7], Holden Maecker[34], Arthur M. Mandelin II[35], Ian Mantel[10,11], Mark Maybury[18,20], Mandy J. McGeachy[14], Joseph Mears[1,5,6,7,8], Alessandra Nerviani[15,16], Dana E. Orange[10,36], Harris Perlman[35], Javier Rangel-Moreno[3], Karim Raza[18,20], Yakir Reshef[1,5,6,7,8], Christopher Ritchlin[3], Felice Rivellese[15,16], William H. Robinson[37], Laurie Rumker[1,5,6,7,8],

**Ilfita Sahbudin**[18,20], **Karen Salomon-Escoto**[38], **Dagmar Scheel-Toellner**[18,20], **Jennifer A. Seifert**[13], **Anvita Singaraju**[10,11], **Melanie H. Smith**[10], **Paul J. Utz**[37], **Kathryn Weinand**[1,5,6,7,8], **Dana Weisenfeld**[1], **Michael H. Weisman**[23,37], **Qian Xiao**[1,5,6,7,8] & **Zhu Zhu**[1]

[21]Accelerating Medicines Partnership Program: Rheumatoid Arthritis and Systemic Lupus Erythematosus (AMP RA/SLE) Network, Bethesda, MD, USA. [22]Division of Rheumatology, Columbia University College of Physicians and Surgeons, New York, NY, USA. [23]Division of Rheumatology, Cedars-Sinai Medical Center, Los Angeles, CA, USA. [24]Department of Pathology and Laboratory Medicine, University of Rochester Medical Center, Rochester, NY, USA. [25]Department of Radiology, University of Pittsburgh Medical Center, Pittsburgh, PA, USA. [26]Department of Pathology and Laboratory Medicine, Hospital for Special Surgery, New York, NY, USA. [27]Division of Allergy, Immunology, and Transplantation, National Institute of Allergy and Infectious Diseases, National Institutes of Health, Bethesda, MD, USA. [28]Northrop Grumman Health Solutions, Rockville, MD, USA. [29]Feinstein Institute for Medical Research, Northwell Health, Manhasset, New York, NY, USA. [30]Department of Arthritis and Clinical Immunology, Oklahoma Medical Research Foundation, Oklahoma City, OK, USA. [31]Division of Clinical Immunology and Rheumatology, Department of Medicine, University of Alabama at Birmingham, Birmingham, AL, USA. [32]Laboratory for Human Immunogenetics, RIKEN Center for Integrative Medical Sciences, Yokohama, Japan. [33]Department of Surgery, Brigham and Women's Hospital and Harvard Medical School, Boston, MA, USA. [34]Human Immune Monitoring Center, Stanford University, Stanford, CA, USA. [35]Division of Rheumatology, Department of Medicine, Northwestern University Feinberg School of Medicine, Chicago, IL, USA. [36]Laboratory of Molecular Neuro-Oncology, The Rockefeller University, New York, NY, USA. [37]Division of Immunology and Rheumatology, Institute for Immunity, Transplantation and Infection, Stanford University School of Medicine, Stanford, CA, USA. [38]Division of Rheumatology, Department of Medicine, University of Massachusetts Chan Medical School, Worcester, MA, USA.

