## [Peer Review File · Nature Communications]

Clonal associations between lymphocyte subsets and functional states in rheumatoid arthritis synoviumREVIEWER COMMENTS

Reviewer #1 (expert in bioinformatic profiling of lymphocytes in rheumatoid arthritis):

The study by Dunlap et al is an ambitious survey of the lymphocyte distribution in rheumatoid synovial tissue. It contains several novel findings but a major critique from this reviewer is the large overlap with data sets from previous studies from the AMP consortia that overshadows the novel aspects. This makes the manuscript challenging to read and appreciate, especially since the figures are too small!

1. The samples are overlapping with previously published studies, hence several figures and analyses have already been done on aliquots of the same samples. This cohort is smaller, and perhaps more homogenous than the previous, but it is the same patients and time points that are studied. Hereby perhaps it is not essential to have both UMAPs and violin plots in every main figure.
2. 1st result paragraph ('Single cell profiling...') it would be interesting to conclude if the assigned pathotypes coincided with level of B cells found in the respective samples.
3. 2nd result paragraph ('Effector CD4+ populations are sign enriched in synovia...'). In contrast to the subheading, overall clusters are discussed without being clearly assigned to peripheral blood or synovial tissue, e.g. line 185 (this large cluster is in fact primarily in blood). Consider changing the heading of this paragraph? Moreover, the authors state (line 189-193) that these results were already found in the previous bigger study (Zhang et al) implicating that some figures are better suited as supplementary information. Lastly, the authors conclude that the different memory/effector populations propagate inflammation (line 201-203). This is an overinterpretation, cells can home and reside in tissue without necessary be drivers of inflammation, and hence not suited for a result section.
4. 3rd result paragraph ('Clonally expanded Tph cells...'). The synovial tissue samples were most likely differing both in size and in number of extracted lymphocytes. Hence some care should be taken when discussing sizes of clones (both T cells and later on B cells) as one is less likely to find clones with lower cell input.
5. 4th result paragraph ('Expanded CD8+...'). Please clarify in figure 3 what is new data and what is already known, perhaps some could be supplementary data to reduce number of items, some figures are very small. Also, the brief mentioning of resident memory CD8+ T

cells in synovia would need to be extended in order to be convincing. Trm are key in barrier tissues, but less established in synovia. This could be discussed.

6. 5th result paragraph ('Predicted virus cross-reactive CD8?...'). I would suggest putting either A or C as suppl to allow the novel data to be better presented. Also, there appears to be duplicated figures e.g. fig 3J = fig S6B and fig 3L=fig S6E. Please double check for potential additional duplications.

7. 6th result paragraph ('B cells exhibit tissue-specific...'). What do the authors mean by tissue-specific? It seems fairly obvious that the cell composition of blood and inflamed tissue are different! Again, the authors conclude (line 422-424) that the found B cells are participants in inflammation. An opposite view would be homing and residency. I.e. this is again an overinterpretation to be in a result section and is better suited in the discussion.

8. The 7th result paragraph ('Accumulation of SHM...'). What would be the rationale to compare mutations between blood and tissue (fig 5F) and are the differences for the two B-naïve clusters biologically meaningful? Same for fig 5G, outcome is expected. Consider moving to supplementary.

9. The 8th result paragraph ('Evidence of in situ antigen exposure...'). What were the yields of plasma blasts and plasma cells and how did they coincide with the pathotypes of the biopsies? Where there plasma cells in all samples?

10. The 9th result paragraph ('Identification of altered T-B cell communication...') What is the rationale to present fig 7A, i.e. co-analyses based on total B cells instead of subsets?

11. In the discussion; the authors have a unique possibility to shed some light on what lymphocytes are found in synovial fluid versus synovial tissue (line 560). Either by analysing other data sets (e.g. PMID: 35831277) in parallel or at least discuss similarities and differences.

12. Citrulline autoimmunity is mentioned a few times, could the authors also speculate where and if RF+ B cells could be residing in the rheumatoid synovia?

13. The cohort is diverse with regard to disease duration and treatment. This could be discussed.

Minor points

1. Line 102, perhaps acknowledge that BCR's mutate and hence their progeny is not identical...

2. Row 221, is average = mean, should it not be median?

3. Consider changing layout of fig 6H and put shared clones in a different color?
4. Line 531, should CXCL be CXCL13?
5. Line 573, which cohorts are being referred to, please clarify.
6. The NK cells is according to lines 343-344 a contamination, and as such should probably not be discussed (line 595) as data cannot be expected to be fully representative.

Reviewer #2 (expert in peripheral T cells in rheumatoid arthritis):

Dunlap et al. performed 5' scRNA-seq and scVDJ-seq on T and B cells from synovial tissue and peripheral blood of RA patients. Certainly important as "resources", a landscape of T and B cell fractions of synovial tissue was obtained. However, the days when superficial analysis of scRNA-seq data was sufficient are over, and these analyses are required to answer immunological and rheumatological questions. In this sense, the style of this manuscript, which lists the results of scRNA-seq analysis and proceeds to the next section without stating the conclusion of each section, is undesirable. As a result, the impact of this manuscript on immunological and rheumatological findings is small in the current situation due to this style and the lack of new immunological and rheumatological conclusions.

First, there are some technical ambiguities in this manuscript.

1. To rigorously analyze clonotypes, information on both TCRa and TCRb sequences for T cells or both H and L chain sequences for B cells is essential. On the other hand, the paper does not specify whether the target cells in this paper are limited to only those in which both sequences are known. Please specify this point. In addition, if this paper includes cells for which only TCRb or only H chain is identified, please analyze only cells for which both sequences are evident.
2. It is not specified whether both TCRa and TCRb are analyzed in pairs for virus-specific sequences. It would also be preferable for the reader to provide the obtained TCR and HLA information in the supplementary.
3. As the author argues, BCRs are subject to somatic mutations. Therefore, 96.5% is too high

a similarity threshold for determining identical clones. Usually, 85% to 80% is used (Soto et al. Nature 2019;566:398). We believe that this study should be analyzed following these thresholds. Also, it is not stated whether the comparison of CDR3 is made with amino acids or nucleic acids.

4. The separation of CD4 and CD8 is generally difficult because the detection sensitivity of CD4 in scRNA-seq is not very good, and CD4 and CD8 T cells that express cytotoxic molecules are sometimes classified into the same cluster. Therefore, their separation is generally tricky. In fact, in Fig. 1B, the entire T cells are clustered as a single population. Furthermore, in this study for clonotyping, it is essential to strictly separate CD8 T cells from CD4 T cells because CD8 T cells tend to proliferate oligo clonally compared to CD4 T cells. Therefore, the strategy to separate CD4 and CD8 should be shown in a supplementary figure like the gating strategy of FACS. Alternatively, separation using totalseq CD4 and CD8, as described in the supplemental table, seems reasonable.

Immunological and rheumatological points

5. TCR clonotype analysis between peripheral blood and joint tissues is insufficient. The study of circulating Tph/Tfh cells is frequently performed in the clinical research of various diseases. Therefore, it is an essential question in rheumatology and human immunology to what extent the dominant clones of CD4 subsets such as Tph and Tfh in RA synovium, maintain the same traits in blood. The authors can analyze T cell clones as Fig. 6I in B cells.

6. Strangely, virus-specific sequencing is only performed on T cells in the synovium. Because of this, the significance of the virus-specific CD8 found in RA synovium in RA is not explained. It is not surprising that systemic oligoclonal expansion of virus-specific CD8 T cells is observed in previously infected individuals with the viruses. The question is to what extent the ratio varies between synovium and peripheral blood. If they are about the same, there would be little association between virus-specific CD8 and arthritis. If they are increased in the synovium, it would suggest some association with arthritis. If an accumulation of virus-specific CD8 in the rheumatoid synovium is observed, the same

analysis should be performed for the TCR of CD4 since the association of virus-specific CD4-positive T cells with rheumatologic pathology is also suggested.

7. As the author has shown, the higher SHM in synovial BCR than in peripheral blood indicates that active B-cell differentiation occurs in the synovium. On the other hand, inflammation-associated B-cell differentiation has been reported to have lower SHM than normal B-cell differentiation (Uzzan et al. Nat Med. 2022;28:766). It is an important question whether SHM occurring in the RA synovium is higher or lower than in the tonsils, etc. Please discuss by comparing with public data or previous reports.

Reviewer #3 (expert in antigen receptor repertoire sequencing):

In the submitted manuscript, the authors undertake a comprehensive single-cell analysis of immune repertoire and cell-cell interactions within the rheumatoid arthritis (RA) synovium, unraveling intricate insights into the disease's immune landscape. The research delves into the potential cross-reactivity of T cells, the contribution of innate T cell populations, and the roles of various B cell subsets in RA pathogenesis.

While the manuscript is meticulously organized and presents impactful findings, I recommend addressing the following points for minor editing and enhancing the manuscript:

1. Cohort Characterization: include disease severity, duration, and treatment history, to provide a comprehensive context for the observed immune signatures.
2. Validation of T Cell Cross-Reactivity: Consider discussing the potential for experimental validation techniques, such as tetramer staining, to solidify the implications of cross-reactive T cell clones.
3. Therapeutic Implications: Expand on the potential therapeutic implications arising from the identified immune interactions and cell states, highlighting their relevance to targeted interventions in RA.

4. How pooling of the cells across patients affects identified subsets?

In conclusion, this manuscript presents significant findings that advance our understanding of the complex immune landscape in RA synovium. With minor editing to address the points highlighted above, I recommend this manuscript for publication.

To the editors and reviewers:

We thank the reviewers for their careful consideration of our manuscript entitled 'Clonal associations of lymphocyte subsets and functional states in rheumatoid arthritis synovium'. We appreciate the positive critiques including the comprehensive nature of the work and its value as a meticulous resource that provides impactful insights into RA disease pathogenesis and the immune landscape in the synovium. We welcome the opportunity to respond to a number of criticisms with additional analysis of our data and better highlighting the novel aspects of the work. Below is a point-by-point response.

Reviewer #1 (expert in bioinformatic profiling of lymphocytes in rheumatoid arthritis):

The study by Dunlap et al is an ambitious survey of the lymphocyte distribution in rheumatoid synovial tissue. It contains several novel findings but a major critique from this reviewer is the large overlap with data sets from previous studies from the AMP consortia that overshadows the novel aspects. This makes the manuscript challenging to read and appreciate, especially since the figures are too small!

1. The samples are overlapping with previously published studies, hence several figures and analyses have already been done on aliquots of the same samples. This cohort is smaller, and perhaps more homogenous than the previous, but it is the same patients and time points that are studied. Hereby perhaps it is not essential to have both UMAPs and violin plots in every main figure.

Response 1: We appreciate the reviewer's input and have moved several plots from the main figures to supplement to streamline the figures (prior Fig 2C, 2D, 3C, 3D, 3K, 4B, 6E).

A novel aspect of our current study is that paired synovial and blood cells were sequenced and analyzed together, whereas the main AMP RA resource paper included only synovium. Thus, the cell clusters that are enumerated here represent an integration between these two compartments with unique UMAPs. Importantly, this allows us to visualize the cell populations that are enriched in the synovium. The analysis of TCR/BCR repertoires, also not included in the AMP RA resource paper, has enabled detection of clones, clonal expansion, and compartment trafficking linked with specific, transcriptomically defined cell states. We also have highlighted the novel findings of the present manuscript below and in a new schematic Figure 8.

- Identification of the most clonally expanded T cell clusters in RA synovium, which includes distinct populations of GMZB CD8, CD4 CTL, CCL5+ CD4 cells, and Tph cells.
- Linked analysis of repertoire and transcriptome highlights Tph cells as selectively enriched in both oligoclonality and transcriptomic signals of TCR activation.
- Definitive demonstration and quantification of multiple innate T cell populations in RA synovium including MAIT cells, made possible by utilization of TCR repertoire data.
- Assessment of predicted viral-specific T cells from RA synovium compared to blood.
- Identification of highly enriched T and B cell subsets in RA synovium compared to blood of same patients.
- Enrichment of multiple functionally distinct activated B cell subsets in the synovium with molecular evidence of selection (higher somatic hypermutation compared to blood counterparts and class switch recombination).
- Demonstration of a developmental connection between ABCs, many of which have a memory phenotype, and synovial plasma cells, with Tph cells and IFN as important predicted drivers of activation and selection in the tissue.

2. 1st result paragraph ('Single cell profiling...') it would be interesting to conclude if the assigned pathotypes coincided with level of B cells found in the respective samples.

Response 2: We agree that it would be interesting to connect B cell accumulation or clonality to pathotypes; however, the sample size in this study is underpowered to examine correlations between assigned pathotypes, CTAPs, and lymphocyte abundance. To aid with the interpretation, we have added data on T cell and B cell proportions at the individual patient level to Supplementary Fig. 1 as new Supplementary Fig. 1C&D.

3. 2nd result paragraph ('Effector CD4+ populations are sign enriched in synovia...'). I contrast to the subheading, overall clusters are discussed without being clearly assigned to peripheral blood or synovial tissue, e.g. line 185 (this large cluster is in fact primarily in blood). Consider changing the heading of this paragraph? Moreover, the authors state (line 189-193) that these results were already found in the previous bigger study (Zhang et al) implicating that some figures are better suited as supplementary information. Lastly, the authors conclude that the different memory/effector populations propagate inflammation (line 201-203). This is an overinterpretation, cells can home and reside in tissue without necessary be drivers of inflammation, and hence not suited for a result section.

Response 3: We appreciate the reviewer's point and have modified the title of the subsection to "Differential abundances of CD4+ T cell populations in synovial tissue and blood."

Regarding the relationship of these analyses to the main AMP Resource study, we intended to indicate that the clusters identified in this repertoire dataset correspond well to clusters in the AMP Resource study, which aids with interpreting cell clusters across the two studies. We would emphasize that this report is unique in its inclusion of paired blood samples, which are not a part of the AMP Resource paper. We agree that the description of the clusters can be streamlined, thus we have moved prior Fig 2B, 2C to the supplement as suggested.

We appreciate the reviewer's point about the concluding statement and have modified it to "Thus, a range of memory/effector T cell populations with distinct transcriptomic signatures are enriched in RA synovium compared to blood."

4. 3rd result paragraph ('Clonally expanded Tph cells...'). The synovial tissue samples were most likely differing both in size and in number of extracted lymphocytes. Hence some care should be taken when discussing sizes of clones (both T cells and later on B cells) as one is less likely to find clones with lower cell input.

Response 4: We agree with the reviewer's point and have added this consideration to this results paragraph: "Though detected clone size can be impacted by the number of cells analyzed, it is notable that the GNLY+ cluster was of lower abundance compared to other CD4 clusters (Figure 2B), and clonal expansion was still detectable."

We have also noted this consideration in the Discussion: "We acknowledge that a potential limitation of our approach is variability in the number of synovial T cells (and B cells) isolated and analyzed from different samples which could bias the clone sizes detected."

In addition, we have clarified in the methods how our approach to sorting and sequencing was performed for each sample in order to avoid bias driven by numbers of lymphocytes analyzed.

Even with this approach, the ultimate number of cells recovered for gene expression and receptor repertoire analysis did vary across synovial samples (Fig 1C, E, G).

5. 4th result paragraph ('Expanded CD8+...'). Please clarify in figure 3 what is new data and what is already known, perhaps some could be supplementary data to reduce number of items, some figures are very small. Also, the brief mentioning of resident memory CD8+ T cells in synovia would need to be extended in order to be convincing. Trm are key in barrier tissues, but less established in synovia. This could be discussed.

Response 5: We appreciate the opportunity to clarify the novelty here. As the overall clustering of CD8 T cells is similar to recent published data; we have moved the violin plot (prior Fig 3C) to the supplement as suggested. The detailed comparative assessment of the phenotypes of the largest CD8 clones in synovium vs blood is new and provides a substantial advance in understanding the relationship between large clones in synovium and blood. Further, the assessment of potential viral-reactive clones has not been previously at this resolution or scale, with a detailed delineation of the phenotypes of both T cells with perfect-match and motif-matched TCRs of viral-reactive T cells.

We agree that the presence of Trm in synovium has been uncertain, yet the recent report from Chang et al, Cell Reports 2021, PMID 34706228 makes a convincing case that Trm accumulate in murine synovium in autoimmune arthritis and contribute to recurrent flares, with some supporting analyses of synovial tissue from RA patients. The cluster that we labeled as 'Trm' shows a strong enrichment of a Trm-associated gene list, which is unique to this cluster. In the context of the Chang et al report, we think it is reasonable to annotate this small cluster as Trm. The cluster in our dataset is quite small, thus we are not able to evaluate these cells in further detail. We have modified the results sentence and also noted the context of the Chang et al report: "We also isolated a population of likely resident memory CD8+ T (Trm) cells, characterized by increased *ZNF683* and *XCL1*, which was supported more broadly through examination of a previously-published Trm gene list and consistent with a recent description of synovial Trm cells."

6. 5th result paragraph ('Predicted virus cross-reactive CD8?...'). I would suggest putting either A or C as suppl to allow the novel data to be better presented. Also, there appears to be duplicated figures e.g. fig 3J = fig S6B and fig 3L=fig S6E. Please double check for potential additional duplications.

Response 6: To improve the presentation of novel data, we have moved prior Fig 3C, 3D to the supplement. Prior figures 3J and 3L are not duplications of prior figure S6B, S6E but depict two different metrics of the analysis (main figure plots show %; supplement figures show absolute numbers of cells and clones). We think that both are valuable to show.

7. 6th result paragraph ('B cells exhibit tissue-specific...'). What do the authors mean by tissue-specific? It seems fairly obvious that the cell composition of blood and inflamed tissue are different! Again, the authors conclude (line 422-424) that the found B cells are participants in inflammation. An opposite view would be homing and residency. I.e. this is again an overinterpretation to be in a result section and is better suited in the discussion.

Response 7: We agree with the reviewer that it is not necessarily unexpected that the B cell composition in blood and tissue are different. However, this is one of the first times that differences in B cell state composition have been specifically delineated between blood and

synovium. We acknowledge that these differences could be driven by either differential homing vs. generation in synovial immune reactions. We have eliminated the sentence ‘The increased proportions of activated B cell populations in synovium support the active participation of these cells in synovial immune responses’ and added more nuanced comments to the discussion.

8. The 7th result paragraph (‘Accumulation of SHM...’). What would be the rationale to compare mutations between blood and tissue (fig 5F) and are the differences for the two B-naïve clusters biologically meaningful? Same for fig 5G, outcome is expected. Consider moving to supplementary.

Response 8: Although SHM has been explored for B cells in blood in various contexts, mutation rates in specific B cell subsets from the synovium are much less clear. Comparison to blood is an important reference point for SHM rates of synovial B cells. The higher mutation rates in corresponding synovial B cell subsets suggests that these cells are under different selection pressures. As discussed above, it is possible that more activated, somatically mutated B cells preferentially home to the synovium, but we favor the hypothesis that SHM occurs in situ. Indeed, new analysis of CDR3 homology at a lower threshold (see below in response to reviewer #2), reveals clonal trees with accumulation of mutations in the synovium. In terms of the differences between SHM rates in the two B-naïve clusters, we do think this is biologically meaningful and suggests that the IgD low naive B cells have been activated and begin to undergo SHM. In Figure 5G, the finding that activated B cells and ABCs in the synovium accumulate SHM has not been previously demonstrated and is thus important to report in the main figures.

9. The 8th result paragraph (‘Evidence of in situ antigen exposure...’). What were the yields of plasma blasts and plasma cells and how did they coincide with the pathotypes of the biopsies? Were there plasma cells in all samples?

Response 9: As shown in Supplementary Fig. 8A, there are plasmablasts and plasma cells in all the samples (except RA07_SYN). Most of the samples are lymphoid pathotype except RA_07 which is pauci-immune and a Fibroid CTAP.

10. The 9th result paragraph (‘Identification of altered T-B cell communication...’) What is the rationale to present fig 7A, i.e. co-analyses based on total B cells instead of subsets?

Response 10: We agree with the reviewer that it is of interest to explore T-B cell interactions at the B cell subset level. We presented these analyses in different combinations in Figure 7 because presenting results of all T cell subsets with all B cell subsets becomes very large and hard to visualize; therefore, we parsed the results into 3 depictions. First, we asked about interactions of different T cell subsets with B cells in total (Fig. 7A). We recognize that this does not capture B cell subset-specific interactions; however, it enables a granular depiction of the number of predicted interactions. We then summarize interactions across all T cell subsets and B cell subsets in heatmap form to condense the visualization (Fig. 7B), which highlighted differences in Tph vs Tfh/Tph interactions in the context of many other interaction pairs. We then show a focused comparison of Tph vs Tfh/Tph interactions with all B cell subsets in both blood and synovium. We feel that this progression adequately balances the need to show tangible examples of the output (Fig. 7A) and also summarized results across the many potential combinations (Fig. 7B).

11. In the discussion; the authors have a unique possibility to shed some light on what

lymphocytes are found in synovial fluid versus synovial tissue (line 560). Either by analysing other data sets (e.g. PMID: 35831277) in parallel or at least discuss similarities and differences.

Response 11: We appreciate the reviewer's suggestion and very much agree that the similarities between synovial fluid and synovial tissue are of major interest. We have qualitatively compared results from this synovial tissue dataset to the Argyrio dataset and other synovial fluid datasets, and some patterns emerge, in particular the presence of a Tfh-appearing population in synovial tissue that seems less abundant in synovial fluid. We are hesitant to speculate further on differential abundances between tissue and fluid (e.g Tregs or other subsets) - this would be best done using paired fluid/tissue analyses, which we do not have here. We have added a statement to the Discussion noting the point above regarding Tph and Tfh cells: "Analyses of synovial tissue performed here suggest that Tfh cells, as contained within the Tfh/Tph cluster, may be more abundant in synovial tissue than in synovial fluid, in which PD-1hi CD4 T cells are predominantly CXCR5- Tph cells."

12. Citrulline autoimmunity is mentioned a few times, could the authors also speculate where and if RF+ B cells could be residing in the rheumatoid synovia?

Response 12: We agree that antigen-specificity is important, though we do not have data on this. We have added consideration of this point to the discussion.

13. The cohort is diverse with regard to disease duration and treatment. This could be discussed.

Response 13: The reviewer is correct that the cohort here is clinically diverse. We have expanded note of this limitation in the discussion.

Minor points

1. Line 102, perhaps acknowledge that BCR's mutate and hence their progeny is not identical...

We have clarified this point.

2. Row 221, is average = mean, should it not be median?

We have confirmed that the mean (average) is shown.

3. Consider changing layout of fig 6H and put shared clones in a different color?

We have modified now figure 6G to highlight shared clones in black.

4. Line 531, should CXCL be CXCL13?

We agree this is somewhat ambiguous. The pathway family identified is 'CXCL', though the factor that drives this association is indeed CXCL13. Since we are listing the 'signaling pathway families' from the analysis output in this sentence, we prefer to maintain the term 'CXCL' but have modified it to 'CXCL chemokines' to reduce the ambiguity.

5. Line 573, which cohorts are being referred to, please clarify.

Thank you for noting this ambiguity. We have stated the point more explicitly: "A striking finding through the current and prior analyses was the near-absence of clonal overlap between GZMK+ cells and GZMB+ cells across tissues, suggesting that these GZMK+ cells do not arrive at the synovium as GZMB-expressing cytotoxic cells..."

6. The NK cells is according to lines 343-344 a contamination, and as such should probably not be discussed (line 595) as data cannot be expected to be fully representative.

We agree and have removed mention of NK cells from the Discussion.

Reviewer #2 (expert in peripheral T cells in rheumatoid arthritis):

Dunlap et al. performed 5' scRNA-seq and scVDJ-seq on T and B cells from synovial tissue and peripheral blood of RA patients. Certainly important as "resources", a landscape of T and B cell fractions of synovial tissue was obtained. However, the days when superficial analysis of scRNA-seq data was sufficient are over, and these analyses are required to answer immunological and rheumatological questions. In this sense, the style of this manuscript, which lists the results of scRNA-seq analysis and proceeds to the next section without stating the conclusion of each section, is undesirable. As a result, the impact of this manuscript on immunological and rheumatological findings is small in the current situation due to this style and the lack of new immunological and rheumatological conclusions.

First, there are some technical ambiguities in this manuscript.

1. To rigorously analyze clonotypes, information on both TCRA and TCRB sequences for T cells or both H and L chain sequences for B cells is essential. On the other hand, the paper does not specify whether the target cells in this paper are limited to only those in which both sequences are known. Please specify this point. In addition, if this paper includes cells for which only TCRB or only H chain is identified, please analyze only cells for which both sequences are evident.

Response 1: As now clarified in the revised manuscript, for the primary TCR analyses presented, cells with both TCRA and TCRB sequences available were included. For the single cell BCR profiling, we only included B cells with paired transcriptomic data and BCR data but did not require both H and L chain sequences. This is because restricting to cells with paired transcriptomic data, H chain, and L chain sequences reduced the number of B cells available for analysis from over 30,000 to less than 15,000. This limited our ability to connect identical or related clones across cell states. Further, published literature supports the use of IgH gene sequencing for definition of clonotypes (Hershberg and Prak, Phil. Trans. R. Soc. B 370: 20140239; Uzzan et al. Nat Med. 2022;28:766; Alamyar et al. Methods Mol. Biol. 2012 882: 569). We did repeat the clonotype analysis in Figure 6 limiting to cells with both H and L chain sequences available, and though the total number of clones identified was smaller (given the smaller cell pool), the data was otherwise similar. For example, trafficking clones reduced from 9 to 5 with similar patterns of transition from one state in the blood to a different state in the synovium.

2. It is not specified whether both TCRA and TCRB are analyzed in pairs for virus-specific sequences. It would also be preferable for the reader to provide the obtained TCR and

HLA information in the supplementary.

Response 2: We agree and have added a detailed description in the methods of this analysis. We used publicly available datasets (VDJdb, McPAS) to obtain TCRs of viral-specific T cells; the majority of data in these datasets are from bulk TCR sequencing with only the beta chain available; therefore, we performed the viral-reactivity analyses using beta chain data only. For exact matching, a 'match' was considered when the CDR3 region, as well as the MHC allele, were the same. For GLIPH2 analysis, the same lists of EBV, CMV, and FLU reactive cells were input alongside patient CD8 TCRs as input. Results were filtered to retain GLIPH groups with previously-identified TCRs presented on an MHC that matched an HLA allele from the patient. GLIPH group results were also filtered for stringency, with a Fisher score < 0.01 . We agree that it is helpful to share both the TCR data and the HLA information; the TCR sequence data will be shared along with all of the other processed sc data. We have now also added the imputed HLA data as Supplementary Table 5.

3. As the author argues, BCRs are subject to somatic mutations. Therefore, 96.5% is too high a similarity threshold for determining identical clones. Usually, 85% to 80% is used (Soto et al. Nature 2019;566:398). We believe that this study should be analyzed following these thresholds. Also, it is not stated whether the comparison of CDR3 is made with amino acids or nucleic acids.

Response 3: The reviewer makes the very important point that clonal identification under different thresholds of CDR3 and VH sequence identity can yield different outcomes. In the original analysis we decided to employ a very stringent 96.5% sequence homology (based on nucleic acid) to irrefutably identify clones and thus establish developmental relationships between cell states, which has not been systematically done before for synovial B cells. In this way we were able to identify cells that started as an ABC or activated B cell state in the synovium and then differentiated to plasmablast or plasma cell. Additionally, we were able to identify identical clones that trafficked from the blood into the synovium, including memory phenotype cells, and became activated B cells, ABCs, or plasma cells. This strongly suggests that B cells home to the synovium and participate in synovial immune responses. We have taken the reviewer's suggestion and repeated the clonal analysis with less stringent demands of clonal identity, specifically 80%. These results are included in Supplementary Fig. 12. Though more clones were identified across multiple cell states within the synovium and trafficking between the blood and synovium at 80% compared to 96.5% identity, the same developmental relationships were identified between activated naive, activated B, ABC, and plasma cells.

4. The separation of CD4 and CD8 is generally difficult because the detection sensitivity of CD4 in scRNA-seq is not very good, and CD4 and CD8 T cells that express cytotoxic molecules are sometimes classified into the same cluster. Therefore, their separation is generally tricky. In fact, in Fig. 1B, the entire T cells are clustered as a single population. Furthermore, in this study for clonotyping, it is essential to strictly separate CD8 T cells from CD4 T cells because CD8 T cells tend to proliferate oligo clonally compared to CD4 T cells. Therefore, the strategy to separate CD4 and CD8 should be shown in a supplementary figure like the gating strategy of FACS. Alternatively, separation using totalseq CD4 and CD8, as described in the supplemental table, seems reasonable.

Response 4: We agree that transcriptomes of CD4 and CD8 subsets can overlap (e.g. CD4 CTL), yet our experience is that in scRNA-seq data of T cells from RA synovium, especially data including some CITE-seq staining, CD4 and CD8 T cell subsets can be quite robustly distinguished. This occurred in the main AMP resource report (Zhang et al, Nature 2023), and

also in these analyses. Figure S3B shows a clear separation of CD4+ and CD8+ regions among T cells, consistent with the ability of transcriptomics to distinguish these populations. Correspondingly, the violin plots (now S4B, S5B) show robust expression of CD8 transcript in the CD8 clusters and clear expression of CD4 transcript in the CD4 clusters, including in the GNLY+ cytotoxic CD4 cluster.

In further support of the faithful detection of CD4 and CD8 in this dataset, we see excellent correspondence between CD4/CD8 transcript and CD4/CD8 protein (ADT) staining. Thus, we think the discrimination of CD4 and CD8 T cells is appropriate to allow assessments of clonality and clonal overlap separately for CD4 T cells and CD8 T cells as presented.

Immunological and rheumatological points

5. TCR clonotype analysis between peripheral blood and joint tissues is insufficient. The study of circulating Tph/Tfh cells is frequently performed in the clinical research of various diseases. Therefore, it is an essential question in rheumatology and human immunology to what extent the dominant clones of CD4 subsets such as Tph and Tfh in RA synovium, maintain the same traits in blood. The authors can analyze T cell clones as Fig. 6I in B cells.

We appreciate the reviewer's emphasis on this point, and we have added additional results regarding this question. We can find some clones that are shared between synovial Tph/Tfh cells and cells in blood; they are predominantly represented in the blood Tph/Tfh clusters, suggesting some clonal sharing between synovial Tph/Tfh and blood Tph/Tfh, though the numbers are small. We agree that this is a valuable point to report and have included it as new Supplementary Fig. 4J. We have added this point to the results and discussion.

Results:

"We also identified a small number of T cells from blood with TCRs that matched synovial Tph or Tfh/Tph cells; these blood T cells were most frequently in the blood Tph or Tph/Tfh clusters (**Supplementary Fig. 4J**)."

Discussion:

"Further, our identification of some shared clones between Tph in synovium and in blood, even with small numbers of total T cells analyzed, supports the notion that a portion of Tph cells in blood have both transcriptomic and clonal relationships to Tph cells in synovium."

6. Strangely, virus-specific sequencing is only performed on T cells in the synovium. Because of this, the significance of the virus-specific CD8 found in RA synovium in RA is not explained. It is not surprising that systemic oligoclonal expansion of virus-specific CD8 T cells is observed in previously infected individuals with the viruses. The question is to what extent the ratio varies between synovium and peripheral blood. If they are about the same, there would be little association between virus-specific CD8 and arthritis. If they are increased in the synovium, it would suggest some association with arthritis. If an accumulation of virus-specific CD8 in the rheumatoid synovium is observed, the same analysis should be performed for the TCR of CD4 since the association of virus-specific CD4-positive T cells with rheumatologic pathology is also suggested.

Response 6: We appreciate the reviewer's enthusiasm about this question, and we have now added an assessment of the predicted virus-specific CD8 T cells in blood (Fig 3J, Supplementary Fig. 6E, 6F, 6J). Taken together, these plots indicate that the number or proportion of virus-reactive cells in the blood is comparable to that in synovium, indicating no particular enrichment of virus-reactive cells in synovium. Analogous to what we observed with synovial T cells, virus-specific CD8 cells from blood show a range of phenotypes that resembles the overall distribution of blood CD8 T cells. We have added these new analyses to the results section:

“For comparison, analysis of CD8 T cells from blood yielded similar results, with a comparable number of viral-reactive T cells, and a similarly broad distribution of cell phenotypes represented by viral-reactive cells (**Supplementary Fig. 5F, 6E&F**).”

“Similar to results with exact matches, a breakdown of the cluster makeup of motif-matching cells and non-matching cells across tissue sources again revealed no significant differences among synovial CD8 T cells, with similar results also obtained for CD8 T cells from blood (**Figure 3J**).”

7. As the author has shown, the higher SHM in synovial BCR than in peripheral blood indicates that active B-cell differentiation occurs in the synovium. On the other hand, inflammation-associated B-cell differentiation has been reported to have lower SHM than normal B-cell differentiation (Uzzan et al. Nat Med. 2022;28:766). It is an important question whether SHM occurring in the RA synovium is higher or lower than in the tonsils, etc. Please discuss by comparing with public data or previous reports.

Response 7: Similar to other inflamed tissues, the SHM rates are lower in the synovium than in tonsil. We have added this important point and reference noted above to the discussion.

Reviewer #3 (expert in antigen receptor repertoire sequencing):

In the submitted manuscript, the authors undertake a comprehensive single-cell analysis of immune repertoire and cell-cell interactions within the rheumatoid arthritis (RA) synovium, unraveling intricate insights into the disease's immune landscape. The research delves into the potential cross-reactivity of T cells, the contribution of innate T cell populations, and the roles of various B cell subsets in RA pathogenesis.

While the manuscript is meticulously organized and presents impactful findings, I

recommend addressing the following points for minor editing and enhancing the manuscript.

1. Cohort Characterization: include disease severity, duration, and treatment history, to provide a comprehensive context for the observed immune signatures.

Response 1: Summarizing the clinical data for RA patients is always a challenge. We elected to provide clinical data on the individual patients in figure form in Supplementary Fig. 1A, indicating age, sex, RA duration, seropositivity, disease activity, treatment history, pathotype, and CTAP.

2. Validation of T Cell Cross-Reactivity: Consider discussing the potential for experimental validation techniques, such as tetramer staining, to solidify the implications of cross-reactive T cell clones.

Response 2: We agree that validation of cross-reactive T cells such as viral-reactive T cells with tetramer staining would be of interest in future studies and have expanded this point to the discussion. "Future work to better define the viral-reactive capacity of T cells within the joint may rely on isolating viral-specific cells using viral peptides bound to tetramers and evaluating them for cross-reactivity to synovial antigens."

3. Therapeutic Implications: Expand on the potential therapeutic implications arising from the identified immune interactions and cell states, highlighting their relevance to targeted interventions in RA.

Response 3: We agree and have expanded on this point in the discussion: "The work further highlights specific lymphocyte populations, including Tph cells, ABC, and activated B cells, that show both transcriptomic signatures of antigen activation and clonal expansion, marking these cell populations as promising therapeutic targets that might be selectively targeted to blunt the pathologic adaptive immune response in RA."

4. How pooling of the cells across patients affects identified subsets?

Response 4: We agree that pooling cells from all patients will result in different clustering than clustering cells separately from each individual patient, yet we think that pooling the data is the most efficient way to generate interpretable results for the cohort. Reassuringly, the clusters from this dataset correspond well to clusters from the AMP RA resource report (Nature 2023), and the clusters each contain cells from most or all patients.

In conclusion, this manuscript presents significant findings that advance our understanding of the complex immune landscape in RA synovium. With minor editing to address the points highlighted above, I recommend this manuscript for publication.

We thank the reviewer for their positive comments and support of this manuscript.

REVIEWERS' COMMENTS

Reviewer #1

This reviewer was no longer available for review and was therefore replaced by Reviewer #2.

Reviewer #2 (Remarks to the Author):

Dunlap et al. generally responded appropriately to the items pointed out by the referee. On the other hand, there are some minor points that need to be corrected.

1. Lu et al. 2018 (Line 101) is not cited.
2. The analysis in this paper does not distinguish between Tph and Tfh cells within the Tph/Tfh cluster. Therefore, no data are presented to support the arrows indicating the clonal relationship between Tfh and Tph cells shown in the newly added Fig. 8. It would be appropriate to discuss the possibility of a relationship between synovial Tph and Tfh cell clones in the discussion section, touching on the limitation of this study and on previous literature showing the clonal relationship between Tph and Tfh cells in LNs of HIV patients. Also, the figure legend of Fig. 8E, "Synovial Tfh cells and Tph cells show significant clonal relationships" could be reworded as "Tph cells and Tph/Tfh cells show significant clonal relationships."

Comments on behalf of reviewer 1.

The authors have addressed all of reviewer 1's comments.

Reviewer #3 (Remarks to the Author):

The authors addressed all the comment. I endorse this paper for publication.

We have addressed the remaining comments from Reviewer #2.

Comment 1. *Lu et al. 2018 (Line 101) is not cited.*

Response 1: We have added this reference.

Comment 2. *The analysis in this paper does not distinguish between Tph and Tfh cells within the Tph/Tfh cluster. Therefore, no data are presented to support the arrows indicating the clonal relationship between Tfh and Tph cells shown in the newly added Fig. 8. It would be appropriate to discuss the possibility of a relationship between synovial Tph and Tfh cell clones in the discussion section, touching on the limitation of this study and on previous literature showing the clonal relationship between Tph and Tfh cells in LNs of HIV patients. Also, the figure legend of Fig. 8E, "Synovial Tfh cells and Tph cells show significant clonal relationships" could be reworded as "Tph cells and Tph/Tfh cells show significant clonal relationships."*

Response 2: We have added a new figure panel that separates Tph vs Tfh cells within the Tph/Tfh cluster based on detection of *CXCR5* transcript. This analysis shows clear clonal overlap between *CXCR5*⁺ cells in the Tph/Tfh cluster (unambiguous Tfh cells) and cells in the Tph cluster (unambiguous Tph cells). This plot supports the conclusion in the manuscript and the Figure 8 schematic as drawn. We have added this plot as **Supplementary Fig 4J** and noted this specific point in the Results and Discussion.

Results: "By subdividing the Tfh/Tph cluster into cells with and without detectable *CXCR5* transcript, we identified a set of clones represented in both *CXCR5*⁺ Tfh cells and cells within the Tph cluster (**Supplementary Fig. 4J**)."

Discussion: "Nonetheless, we observed that Tph cells from multiple synovial tissue samples were clonally expanded and related to ***CXCR5*⁺** Tfh cells, consistent with recent work tracking clonal relationships of Tph cells in synovial fluid [29]."

RESPONSE TO REVIEWERS' COMMENTS

Reviewer #2 (Remarks to the Author):

Dunlap et al. generally responded appropriately to the items pointed out by the referee. On the other hand, there are some minor points that need to be corrected.

Comment 1. Lu et al. 2018 (Line 101) is not cited.

Response 1: We have added this reference.

Comment 2. The analysis in this paper does not distinguish between Tph and Tfh cells within the Tph/Tfh cluster. Therefore, no data are presented to support the arrows indicating the clonal relationship between Tfh and Tph cells shown in the newly added Fig. 8. It would be appropriate to discuss the possibility of a relationship between synovial Tph and Tfh cell clones in the discussion section, touching on the limitation of this study and on previous literature showing the clonal relationship between Tph and Tfh cells in LNs of HIV patients. Also, the figure legend of Fig. 8E, "Synovial Tfh cells and Tph cells show significant clonal relationships" could be reworded as "Tph cells and Tph/Tfh cells show significant clonal relationships".

Response 2: We thank the reviewer for raising this point, which we now clarify further. We have added a new figure panel that separates Tph vs Tfh cells within the Tph/Tfh cluster based on detection of *CXCR5* transcript. This analysis shows clear clonal overlap between *CXCR5*⁺ cells in the Tph/Tfh cluster (unambiguous Tfh cells) and cells in the Tph cluster (unambiguous Tph cells). This plot supports the conclusion in the manuscript and the Figure 8 schematic as drawn. We have added this plot as Supplementary Fig 4J and noted this specific point in the Results and Discussion.

Results: "By subdividing the Tfh/Tph cluster into cells with and without detectable *CXCR5* transcript, we identified a set of clones represented in both *CXCR5*⁺ Tfh cells and cells within the Tph cluster (Supplementary Fig. 4J)."

Discussion: "Nonetheless, we observed that Tph cells from multiple synovial tissue samples were clonally expanded and related to *CXCR5*⁺ Tfh cells, consistent with recent work tracking clonal relationships of Tph cells in synovial fluid [29]."

Comments on behalf of Reviewer #1. The authors have addressed all of reviewer 1's comments.

Response 1: We thank the reviewer #2 for considering these points on behalf of Reviewer #1.

Reviewer #3 (Remarks to the Author):

Comment: The authors addressed all the comment. I endorse this paper for publication.

Response: We thank the reviewer for their endorsement.